# All-printed chip-less wearable neuromorphic system for multimodal physicochemical health monitoring

Yongsuk Choi[1,2,4], Peng Jin [1,4], Sanghyun Lee[1,3,4], Yu Song [1], Roland Yingjie Tay[1], Gwangmook Kim [1], Jounghyun Yoo [1], Hong Han[1], Jeonghee Yeom [1,2], Jeong Ho Cho [3] ✉, Dong-Hwan Kim [2] ✉ & Wei Gao [1] ✉

Recent advancements in wearable sensor technologies have enabled real-time monitoring of physiological and biochemical signals, opening new opportunities for personalized healthcare applications. However, conventional wearable devices often depend on rigid electronics components for signal transduction, processing, and wireless communications, leading to compromised signal quality due to the mechanical mismatches with the soft, flexible nature of human skin. Additionally, current computing technologies face substantial challenges in efficiently processing these vast datasets, with limitations in scalability, high power consumption, and a heavy reliance on external internet resources, which also poses security risks. To address these challenges, we have developed a miniaturized, standalone, chip-less wearable neuromorphic system capable of simultaneously monitoring, processing, and analyzing multimodal physicochemical biomarker data (i.e., metabolites, cardiac activities, and core body temperature). By leveraging scalable printing technology, we fabricated artificial synapses that function as both sensors and analog processing units, integrating them alongside printed synaptic nodes into a compact wearable system embedded with a medical diagnostic algorithm for multimodal data processing and decision making. The feasibility of this flexible wearable neuromorphic system was demonstrated in sepsis diagnosis and patient data classification, highlighting the potential of this wearable technology for real-time medical diagnostics.

The rapid advancement of wearable sensors has revolutionized biomedical data collection, enabling the generation of large-scale, unstructured data for personalized healthcare applications such as human activity monitoring, biosignal analysis, and disease progression tracking[1–7]. These devices have the potential to transform healthcare by facilitating continuous, non-invasive monitoring of physiological parameters including heart rate (HR), core body temperature (CBT), and even biomolecular markers, which allows for early detection, prevention, and management of diseases[8–15]. Despite these advancements, current wearable devices often rely on rigid electronic components, which create critical mechanical mismatches with the soft, flexible nature of human skin. These mismatches compromise comfort, signal quality, and long-term usability[16–20]. In addition, these devices predominantly focus on single-functionality sensing and

[1]Andrew and Peggy Cherng Department of Medical Engineering, California Institute of Technology, Pasadena, CA, USA. [2]Department of Chemical Engineering, Sungkyunkwan University, Suwon, Republic of Korea. [3]Department of Chemical and Biomolecular Engineering, Yonsei University, Seoul, Republic of Korea. [4]These authors contributed equally: Yongsuk Choi, Peng Jin, Sanghyun Lee. ✉e-mail: jhcho94@yonsei.ac.kr; dhkim1@skku.edu; weigao@caltech.edu

isolated data stream collection, without the capability to integrate and analyze the dynamic interactions of multimodal physiological signals, limiting their potential for delivering comprehensive health insights[21–25].

The increasing complexity and volume of biomedical data also present substantial challenges. Traditional computing technologies struggle with the scalability needed to process these vast datasets efficiently, often requiring high power consumption and dependence on external internet resources. This reliance introduces risks related to data security and privacy, especially when sensitive biomedical information is transmitted to remote servers[26]. Furthermore, the use of rigid, silicon-based computing components exacerbates issues of mechanical compatibility, further reducing wearability and limiting long-term user comfort[20,27].

Brain-inspired computing technologies, such as neuromorphic systems, offer an exciting avenue for addressing these limitations. By emulating the brain's complex neural architecture, neuromorphic systems enable efficient, large-scale data processing with reduced

power consumption[28–32]. However, their application in wearable technologies has been hindered by substantial resource demands, high fabrication costs, and the complexities of integrating flexible materials and multifunctional devices with advanced computing architectures[29,33,34].

These gaps underscore the strong need for next-generation wearable systems that seamlessly integrate multimodal sensing with advanced cascading data processing capabilities to deliver personalized, real-time healthcare insights. Such systems must overcome current limitations in scalability, power consumption, and mechanical compatibility while providing secure, on-body computing solutions for continuous monitoring and medical decision-making.

In this study, we introduce a chip-less wearable sensor-processor integrated neuromorphic system (CSPINS) that can simultaneously collect, process, and analyze multimodal physicochemical biomarker data in real time for personalized healthcare applications (Fig. 1a). CSPINS features a neuromorphic processing layer composed of arrays of artificial synapses and nodes, forming a lightweight, skin-conformal

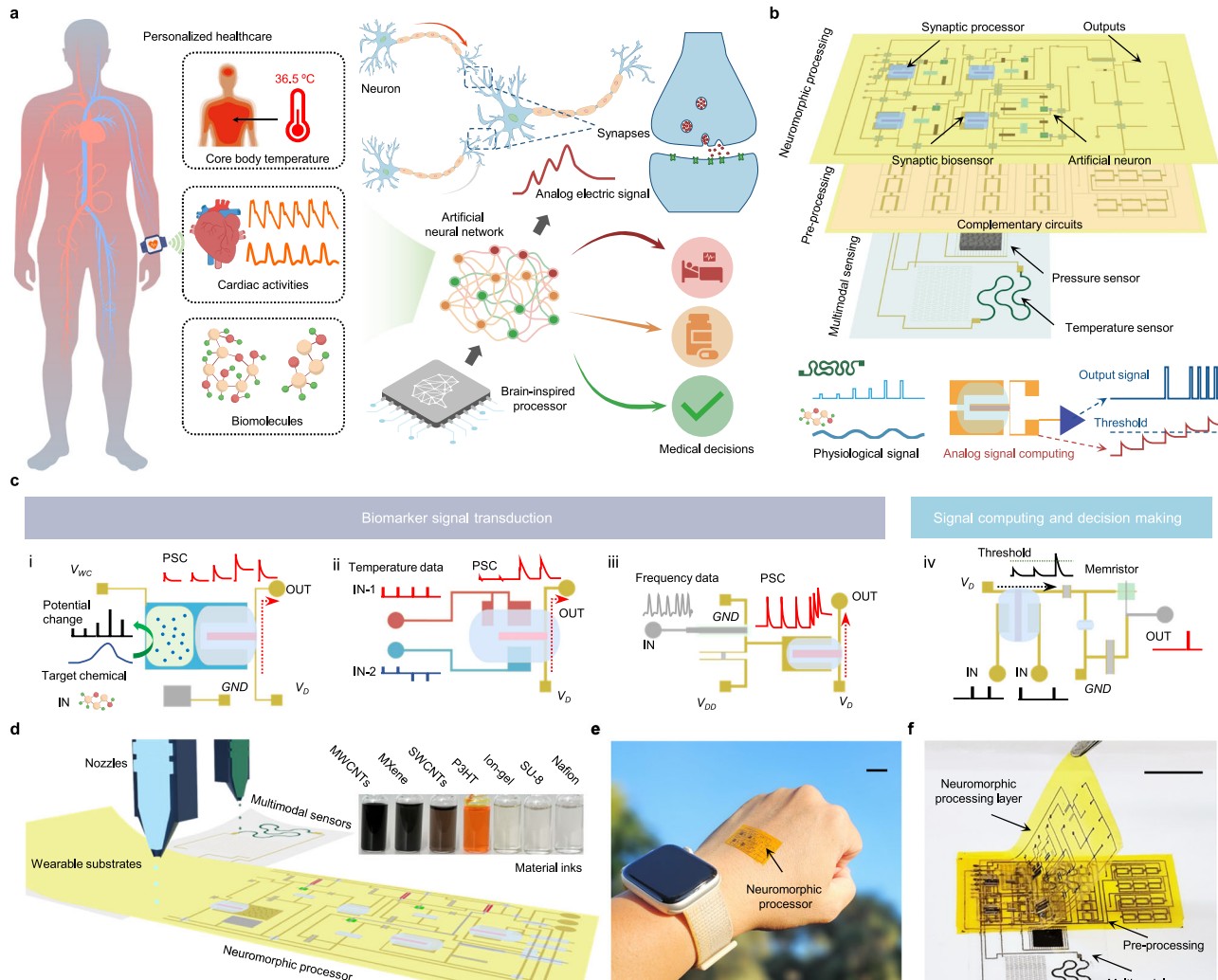

**Fig. 1 | All-printed chip-less wearable sensor-processor integrated neuromorphic system (CSPINS) for multimodal health monitoring. a** Schematic illustrations of the wearable CSPINS, designed to collect, process, and analyze multimodal physicochemical data for real-time medical decision-making. **b** Layered architecture of CSPINS, consisting of a multimodal physical sensing layer, a complementary circuit amplifier layer, and a synaptic sensing and neuromorphic processing layer. **c** Data flow within CSPINS, detailing the stages of signal transduction and processing of synaptic biochemical sensors (i), core body temperature sensor (ii), heart rate sensor (iii), as well as computing and decision-making operations (iv).

IN, input; OUT, output; $GND$, ground; $V_{DD}$, positive supply voltage; $V_D$, drain voltage; $V_{WC}$, weight-control voltage. **d** Scalable fabrication of the wearable neuromorphic device using inkjet printing with a variety of organic and inorganic nanomaterial inks. PI, polyimide; PET, polyethylene terephthalate; MWCNTs, multi-walled carbon nanotubes; SWCNTs, single-walled nanotubes; P3HT, poly(3-hexylthiophene). **e** Optical image of the wearable neuromorphic processor attached on skin. Scale bar, 1 cm. **f** Optical image of the fully assembled CSPINS, featuring integrated physicochemical health sensors and processing units. Scale bar, 1 cm.

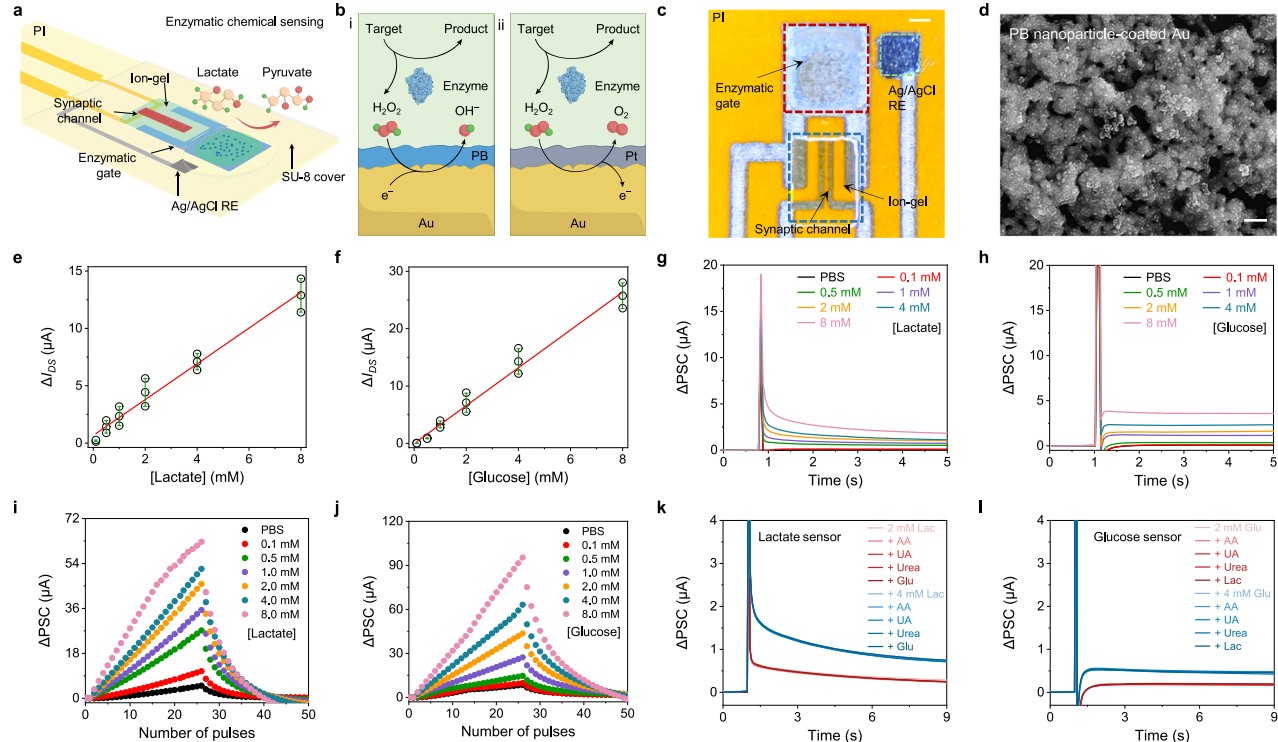

**Fig. 2 | Design and characterization of the synaptic biochemical sensors.**
**a** Schematic design of the enzymatic synaptic biosensors. RE, reference electrode.
**b** Redox reaction mechanism of the synaptic biosensors: oxidases catalyze the oxidation of target molecules, producing $H_2O_2$. This is subsequently catalyzed by Pt or Prussian blue (PB) nanoparticles, resulting in the formation of oxygen or hydroxide radicals, respectively. These reactions facilitate electron donation or withdrawal from Au electrodes, generating synaptic currents. **c** Optical microscopy image of the synaptic biochemical sensor. **d** Scanning electron microscopy (SEM) image of a PB nanoparticle-coated Au electrode. Similar morphological features are observed at more than five locations in each of the three independently prepared samples. Scale bar, 400 nm. **e**, **f** Responses of the lactate (**e**) and glucose (**f**) synaptic sensors in target analyte solutions. Redlines in (**e** and **f**) indicate a linear fit. The

measure of center is represented as the mean, and the error bars indicate the standard deviation (s.d.) from three sensors. **g**, **h** Excitatory postsynaptic current (EPSC) response of the lactate (**g**) and glucose (**h**) synaptic sensors under varied target analyte concentrations. **i**, **j** Long-term potentiation/depression (LTP/D) response of the lactate (**i**) and glucose (**j**) synaptic sensors under varied target analyte concentrations. **k** EPSC response of lactate synaptic sensor in the presence of lactate (Lac) and biochemical interferences, including ascorbic acid (AA) (60 μM), uric acid (UA) (0.3 mM), urea (5 mM), and glucose (Glu) (4 mM). **l** EPSC response of the synaptic glucose sensor in the presence of glucose and biochemical interferences, including ascorbic acid (AA) (60 μM), uric acid (UA) (0.3 mM), urea (5 mM), and lactate (4 mM).

platform that enables on-body data collection, processing, and decision making without reliance on external computing resources (Fig. 1b). By seamlessly integrating multimodal sensing—including HR, CBT, and biochemical analytes—with on-body neuromorphic processing, CSPINS provides autonomous, efficient, and real-time healthcare monitoring and analysis.

## Results and discussion

To achieve efficient analog computation in a miniaturized form, CSPINS employs various signal transduction, processing, and computing strategies (Fig. 1c). The system incorporates synaptic biochemical sensors that directly convert molecular signals into analog currents (synaptic currents), eliminating the need for complex digital circuits and ensuring high-accuracy biochemical detection. A synaptic processor coupled with a dual-sensor design simultaneously measures both skin and device surface temperatures, enabling accurate calculation of CBT using a single-heat flux model[35]. In addition, an on-body HR encoding system converts the mechanical pulse from the skin into analog synaptic currents, providing a robust means for real-time cardiovascular monitoring. The accumulated analog signals from these multimodal sensors are processed through a synaptic node circuit that performs decision-making based on optimized threshold levels and biologically inspired connections. These design features collectively make CSPINS a versatile, autonomous tool for continuous and integrated health monitoring.

The components of CSPINS—including all artificial synapses, memristors, resistors, and capacitors required for the node circuit—are fabricated using scalable inkjet printing technology on thin, flexible polymeric substrates, leveraging customized organic and inorganic ink formulations (Fig. 1d, Supplementary Table 1, and Supplementary Figs. 1 and 2). The fully assembled system, comprising printed synaptic molecular sensors, neuromorphic processors, amplifiers, and 3D-printed physical sensors, is mechanically flexible and adheres well to the skin (Fig. 1e, f and Supplementary Fig. 3). To demonstrate the functionality and transformative potential of CSPINS, we successfully applied it to the diagnosis and classification of sepsis by analyzing clinical data from healthy participants and patients with varying stages of sepsis, highlighting its ability to enable real-time healthcare diagnostics and personalized medical interventions.

## Artificial synapse for biomolecular data collection and processing

A key component of the CSPINS is synaptic biochemical sensors, which directly convert biomolecular marker levels into computable analog signals (Fig. 2a). These sensors utilize a transistor-type synapse that modulates gate potential to induce substantial changes in channel current, generating an analog signal due to the inherent latency between channel and gate[28–30,36]. By functionalizing the gate electrode of a synapse with an enzymatic signal transduction layer, we achieve real-time conversion of biocatalytic reactions into synaptic currents.

The artificial synapse-based sensor employs Prussian blue nanoparticles (PBNPs) or platinum nanoparticles (PtNPs) as the mediator to catalyze the redox reaction of hydrogen peroxide generated in the oxidase-catalyzed reactions (Fig. 2b). PB reduces $H_2O_2$ to hydroxide ions, while Pt oxidizes $H_2O_2$ to produce oxygen. These reactions generate opposite electron flows between the mediator layer and the Au electrode, causing chemically induced potential changes that effectively modulate the synaptic current.

To fabricate the synaptic biochemical sensor, inkjet-printed Au and Ag electrodes, along with interconnects, were patterned on a polyimide (PI) substrate (Supplementary Fig. 4). The synaptic channel, composed of a p-type material combining single-walled carbon nanotube (SWCNT) and poly(3-hexylthiophene) (P3HT), was printed and precisely separated from the gate electrode by plasma etching to remove any unintended residues (Supplementary Fig. 5). The PB or Pt mediator layer and a chitosan/enzyme film were immobilized onto the Au electrode, followed by the printing of a photo cross-linkable ion gel as the gate dielectric (Fig. 2c and Supplementary Fig. 6). Scanning electron microscopy (SEM) confirmed the uniform coverage of PBNPs or PtNPs on the Au surface (Fig. 2d and Supplementary Fig. 7).

The electrical transistor characteristics and synaptic plasticity of the device were validated by applying voltage pulses to the gate terminal (Supplementary Fig. 8). The SWCNT/P3HT heterostructure exhibited stable current characteristics above 0.1 µA at a neutral voltage of 0 V, owing to the high conductivity of SWCNTs. In addition, the inherent ion-permeability of P3HT enabled synaptic properties such as excitatory/inhibitory postsynaptic currents (EPSC/IPSC) and long-term potentiation/depression (LTP/D) under voltage pulses.

Lactate oxidase (LOx) and glucose oxidase (GOx) were functionalized on the sensor to target key molecular biomarkers lactic acid and glucose, respectively. To evaluate sensor performance, phosphate-buffered saline (PBS) solutions with varying target concentrations (0–8 mM) were applied to the functionalized gate while operation pulses were applied (Fig. 2e, f and Supplementary Figs. 9 and 10). The synaptic biosensors showed linear relationships between baseline current and target concentrations at $V_G = 0$ V (Fig. 2g, h). In addition, synaptic current changes in response to target concentrations were recorded, demonstrating that LTP/D characteristics vary proportionally with analyte levels (Fig. 2i, j). Enzymatic reaction-induced potential changes substantially impacted peak and retention currents, proving that the synaptic biochemical sensor can successfully convert and amplify analyte concentrations into analog signals. Furthermore, the sensor showed high selectivity for the target molecule even in the presence of potential biochemical interferences, such as ascorbic acid, uric acid, and urea (Fig. 2k, l and Supplementary Fig. 11), and demonstrated high repeatability and long-term stability through extended testing (Supplementary Fig. 12).

The thickness of the PB layer plays a critical role in determining sensing performance, with thinner PB layers yielding higher sensitivity (Supplementary Fig. 13). In addition, replacing PBNPs with PtNPs and employing n-type semiconductor channel ($In_2O_3$) reversed the electron flow, resulting in an opposite potential change while preserving the positive relation between target concentration and device current. This approach enables the development of high-performance n-type transistor biosensors (Supplementary Fig. 14). These findings demonstrate the versatility of the system, as a wide range of biochemical signals can be targeted and converted into current variations suitable for use in diverse analog processing applications.

## Wearable analog processor for core body temperature monitoring

CBT is a critical physiological parameter that reflects the body's internal thermal state. It plays a fundamental role in maintaining homeostasis and provides valuable insights into physiological processes, overall health status, and potential disease conditions[21,37–39].

However, accurate CBT measurement remains highly challenging for conventional wearable devices due to the variability of skin temperature, which can be substantially lower than CBT and is easily influenced by environmental factors[37–39]. In CSPINS, we addressed this challenge by implementing a CBT mining strategy leverages the analog computational capabilities of artificial synapses (Fig. 3a). CBT is determined using a single-heat flux model[35] that accounts for the temperature gradient between the skin and the surface of the wearable device: $CBT = T_{skin} + (T_{skin} - T_{Al}) \times R_{skin}/R_{device}$, where $T_{skin}$ and $T_{Al}$ represent the interfacial temperatures of the skin-device interface and the device-ambient interface, respectively, and $R_{skin}$ and $R_{device}$ denote the thermal resistances of the skin and the device. To implement this model, we developed a wearable device that integrates two temperature sensors with a synaptic calculator (Fig. 3b).

The temperature sensors were fabricated using inkjet-printed wavy MXene lines, which exhibited substantial resistance changes in response to temperature variations (Fig. 3c, d and Supplementary Fig. 15). These sensors convert temperature into a voltage signal via a voltage divider circuit with thin and long Au lines reference resistors. Two identical temperature sensors, separated by a 2-mm PDMS thermal resistor, were used to measure $T_{skin}$ and $T_{Al}$. The thermal resistance ratio between the skin and the device in the system was measured to be 1.2 (Supplementary Table 2), consistent with known thermal properties of PDMS and skin. For CBT calculation, we employed an inkjet-printed multi-gate ion-gel synapse capable of dynamic weight updates at each gate terminal by modulating the effective gate area (Supplementary Fig. 16).

The MXene line resistance decreased linearly from 8.8 kΩ to 4.4 kΩ as the temperature increased from 20 °C to 50 °C, whereas the Au reference resistor remained stable (Fig. 3e). This resistance variation produced a corresponding voltage output ranging from 3.1 V to 3.9 V (Fig. 3f), which was fed to the gate terminal of the ion-gel synapse to generate an analog thermal signal (Supplementary Fig. 17). Given the narrow temperature range for CBT (30–40 °C), the analog signal was amplified and precisely processed for improved accuracy. The LTP/D curves of the synapse under potentiation and depression pulses (Fig. 3g) achieved up to fivefold amplification through multiple pulses. The relationship between temperature, voltage, and synaptic current was further analyzed under varying pulse conditions, confirming a consistent, linear response (Fig. 3h and Supplementary Fig. 18).

The synapses performed arithmetic operations on synaptic currents by utilizing separate gate electrodes for positively and negatively charged ion movements (Fig. 3i). Additive currents were generated with simultaneous potentiation pulses, while subtractive currents were achieved by combining potentiation and inhibitory pulses. CBT mining was executed by connecting the $T_{skin}$ and $T_{Al}$ nodes to a multi-gate synapse, which implemented the calculation through sequential biosignal input (Fig. 3j). The process involved three initial pulse input to encode the term $(T_{skin} - T_{Al}) \times R_{skin}/R_{device}$, followed by three additional updates to add $T_{skin}$. The computed CBT values showed high consistency across various indoor or outdoor conditions and different body locations. The system used three pulses to achieve sufficient synaptic current changes greater than 10 µA.

The CBT mining device maintained reliable performance during dynamic activities, including transitions between indoor and cold outdoor environments and during light or high-intensity exercises (Fig. 3k). The accuracy of CBT measured from the wearable analog processor showed strong agreement with both mathematically calculated CBT values and medical-grade armpit thermometer readings, confirming the device's robustness and practical utility.

## Wearable analog processor for heart rate encoding

Cardiovascular signals, such as HR, are vital for real-time assessment of heart and circulatory system health, playing a crucial role in the early detection of and response to physiological abnormalities[40]. While many wearable sensors can collect cardiovascular signals, accurately

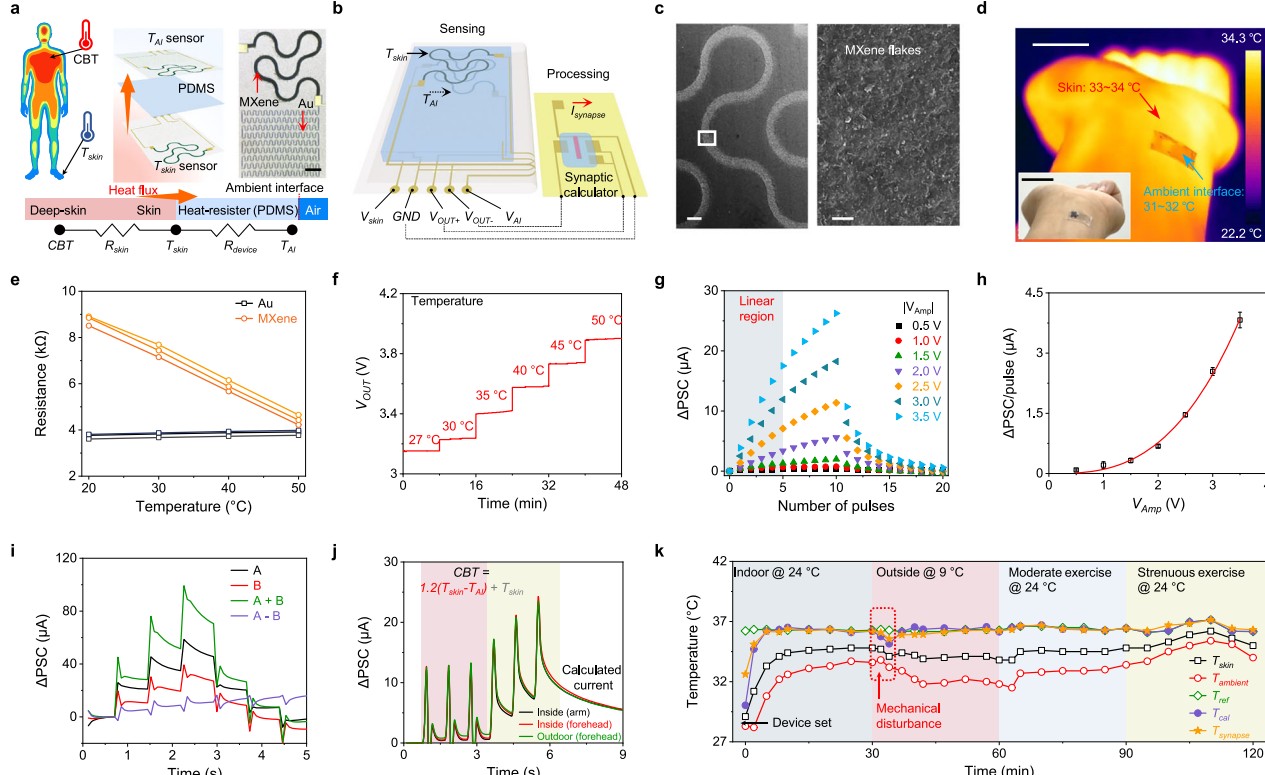

**Fig. 3 | Design and characterization of the wearable analog processor for core-body temperature (CBT) monitoring. a** Schematic of the wearable CBT sensing mechanism based on a single-heat flux model. $T_{skin}$, temperature of skin-device interface; $T_{AI}$, temperature of device-ambient interface; $R_{skin}$ and $R_{device}$, thermal resistances of the skin and the device, respectively. Inset, optical image highlighting key components including Au reference lines and temperature-sensitive MXene lines. Similar morphological features are observed at more than five locations in each of the three independently prepared samples. Scale bar, 200 μm. **b** Detailed schematic of the wearable CBT monitoring system with operational details. Input voltages are set at $V_{skin} = -7.5\,V$ and $V_{AI} = 7.5\,V$. $V_{OUT+}$ and $V_{OUT-}$ represent the synaptic inputs corresponding to $T_{skin}$ and $T_{AI}$. **c** SEM images of the inkjet-printed MXene line surface. Scale bars, 200 μm (Left) and 1 μm (Right). **d** Thermal and optical (inset) images of the wearable CBT monitor attached to human skin. Scale bars, 2 cm. **e** The resistance variation of the Au and MXene lines as a function of temperature. **f** Output voltage from the voltage divider circuit, comprising Au and MXene lines, under varying temperatures with a supply voltage of 7.5 V. **g** LTP/D characteristics of the synapse under different voltage amplitudes. **h** Plots of PSC change per single weight update as a function of the voltage amplitude. $V_{amp}$, voltage amplitude. Redline, exponential growth fitting. Error bars represent s.d. of the mean from 5 synaptic transistors. **i** Additive and subtractive characteristics of the synaptic calculator with two biosignal input terminals. **j** Real-time biosignal input plots obtained from attaching CBT sensors to various body parts in different environments. **k** Dynamic temperature plots recorded over a 2 h period, including $T_{skin}$, $T_{AI}$, mathematically calculated CBT based on sensor readings ($T_{cal}$), and synaptic CBT output ($T_{synapse}$), and CBT values measured using a medical-grade armpit thermometer ($T_{ref}$).

translating their frequency-based information into usable electrical signals without relying on conventional rigid, complex, and high-power-consuming circuits remains highly challenging[1,2]. To address this, we developed a wearable analog processor that amplifies electrical signals generated by pressure-sensitive materials and processes them as synaptic current signals using neuromorphic devices (Fig. 4a).

The periodic oscillations generated by the heartbeats cause resistance changes in the conductive sponge, which are converted into voltage pulses via a voltage divider circuit. These resulting voltage pulses are amplified by complementary circuitry printed on a flexible substrate and subsequently converted into PSCs by an ion-gel-gated synapse. The synapse mimics the millisecond-scale operation of human synapses by inducing rapid ion accumulation or dissociation at the channel and gate electrodes in response to voltage pulses. The measured PSC changes increase proportionally with higher input pulse frequencies (Supplementary Fig. 19).

Heartbeat signals were detected using a 3D-printed conductive sponge-based pressure sensor made from a multi-walled carbon nanotubes/PDMS composite (MWCNT/PDMS) (Fig. 4b, c and Supplementary Fig. 20)[41]. The porous sponge structure exhibits high sensitivity to pressure changes (Fig. 4d and Supplementary Fig. 21), allowing it to convert minor skin deformations from the expansion and contraction of blood vessels into a ~ 5% resistance change (Fig. 4e). To

validate the sensor's accuracy, we simultaneously attached a commercial HR sensor and the conductive sponge to a human subject. The sponge successfully converted heartbeats into resistance and voltage signals corresponding to 80–160 beats per minute (BPM) (Fig. 4f and Supplementary Fig. 22), showing strong agreement with the readings from the commercial HR monitor (Fig. 4g).

A voltage divider circuit, designed with inkjet-printed MWCNTs reference resistors (Supplementary Fig. 23), converted the resistance change into voltage pulses. Optimized with a supply voltage of − 10 V and a reference resistance of 12 kΩ, the circuit ensured compatibility with the sponge's resistance change. A multi-stage amplifier circuit further increased the voltage signal amplitude, enabling reliable PSC generation (Supplementary Figs. 24 and 25). The overall system effectively transformed cardiovascular signals (Fig. 4e) into voltage pulses (Fig. 4f), which were then processed by the synapse (Fig. 4h).

The HR signal processing unit operated on a 10-second data input cycle and a 0.5-second reset period. As HR increased, more pulse signals were delivered to the synapse during the biosignal input window, resulting in higher PSC values (Fig. 4i). A linear relationship between PSC changes and HR was observed across a range of 55–160 BPM (Fig. 4j). To assess real-time performance, the system was tested alongside a commercial HR monitor during rest and intense exercise over a 2-hour period (Fig. 4k). The synaptic PSC changes closely

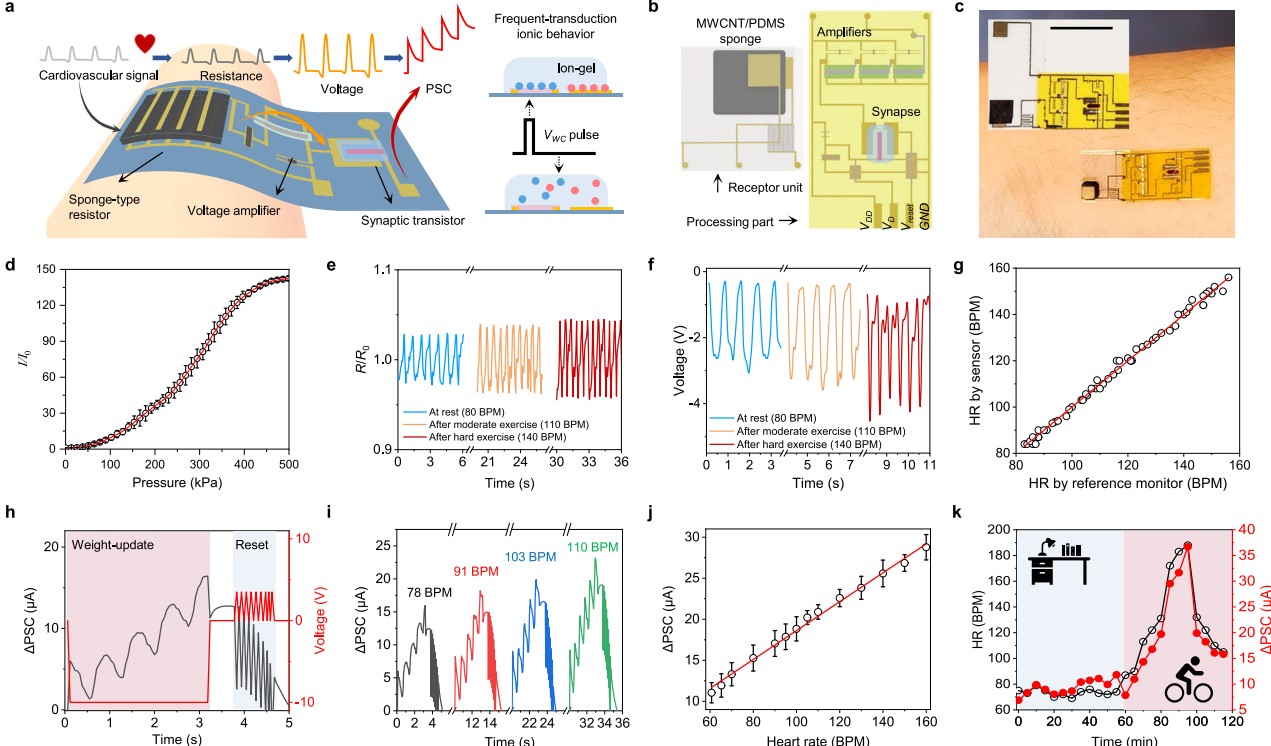

**Fig. 4 | Design and characterization of the wearable processor for heart rate (HR) monitoring. a** Schematic illustration of the wearable HR monitoring system, which translates cardiovascular signals into synaptic currents. **b** Schematics of the HR processing module components, comprising the receptor (pressure-sensitive sensor) and processor (synaptic device). **c** Optical images of a wearable HR processing module, showing the integration of skin-attached receptors and processors. Scale bar, 1 cm. **d** Calibration plot of the conductive sponge-based pressure sensor under varying pressure levels. I and $I_0$ represent the current responses under applied pressure load and baseline (no pressure load), respectively. Error bars represent s.d. of the mean from 5 different samples. $V_D = 1$ V, $I_O \sim 26$ μA. **e** Resistance variations of the conductive sponge under different body conditions. R and $R_0$ represent the resistance under exercise and at rest, respectively. $R_O \sim 38$ kΩ. BPM,

beats per min. **f** Voltage signal output after processing the resistance changes through the amplifier circuit. **g** Calibration curve correlating the frequency of the amplified voltage pulses with a commercial HR monitor. **h** Characterization of the PSC changes during HR module operation, with readings taken after 3 s of biosignal input, followed by a reset. **i** Real-time PSC changes updated under varying body conditions. Inset numbers were collected using a commercial HR monitor. **j** Calibration curve comparing PSC changes from the synaptic HR processor with readings from the commercial HR monitor. Error bars represent s.d. of the mean from 5 repeated measurements. **k** Continuous 2 h HR monitoring with a commercial HR monitor (black) and the wearable synaptic HR processor (red) during various activities performed by the subject.

tracked HR variations, demonstrating the wearable analog processor's ability for real-time cardiovascular signal processing and monitoring.

## Wearable synaptic node for neuromorphic signal processing

Analog signals inherently carry extensive information, yet to enable computation with these signals, devices capable of making binary decisions (e.g., '0' and '1') are essential[42]. In biological systems, the human neural network processes analog signals accumulated in synapses through neurons based on threshold firing properties (Fig. 5a). Mimicking this behavior, the development of the synaptic node is a critical step toward precise data processing of accumulated analog signals and enabling neuromorphic computing.

Here, we present a wearable synaptic node circuit that generates a decision signal when accumulated synaptic currents surpass a threshold, achieved via the on/off switching behavior of a memristor (Fig. 5b). The node circuit, including its memristor, resistors ($R_O$ and $R_1$), and capacitors, was fabricated on a thin PI film via inkjet printing (Supplementary Fig. 26). The memristor, a central component, features a vertical crossbar structure of Au/Nafion/Ag, leveraging Nafion for its excellent mechanical flexibility and compatibility with scalable solution-based printing processing (Supplementary Fig. 27). When the voltage between Au and Ag exceeds the memristor's switching voltage, Ag ions migrate through the proton-conducting Nafion layer, forming

conductive filaments that transition the device from a high-resistance state (HRS) to a low-resistance state (LRS) (Fig. 5c). When the voltage decreases, the filaments dissolve, returning the memristor to HRS.

To ensure optimal charging and discharging behavior of the node circuit, the resistances of inkjet-printed MWCNTs resistors $R_O$ and $R_1$ were tuned to 30 kΩ and 200 kΩ, respectively (Fig. 5d). The memristor's switching voltage was fine-tuned between 0.1 and 16 V by optimizing the Nafion thickness and Ag line width (Fig. 5e, f and Supplementary Figs. 28, 29). The circuit's performance was evaluated by applying 10 Hz input pulses with increasing amplitude (0–20 V) and measuring the output signal (Fig. 5g, h). As the capacitor accumulated charge, the voltage increased until the memristor's switching threshold was reached, at which point the memristor transitions to LRS, causing a rapid voltage to increase at the output terminal (Supplementary Fig. 30). The threshold voltage of synaptic node closely matched the memristor's switching voltage, showing a voltage increase exceeding 10 × upon activation.

To demonstrate analog computing capabilities, we integrated the synaptic device with node circuits set to have distinct threshold levels. A set of read voltages (1 V) and potentiation/depression voltage pulses (±2.5 V) were applied to the synapse's gate electrode (Fig. 5i). The synapse conductance varied between 0.2 to 27.1 mS, with the threshold levels of low-set node and high-set node estimated at 2.3 and 3.1 mS, respectively (Fig. 5j). The corresponding output signals of the

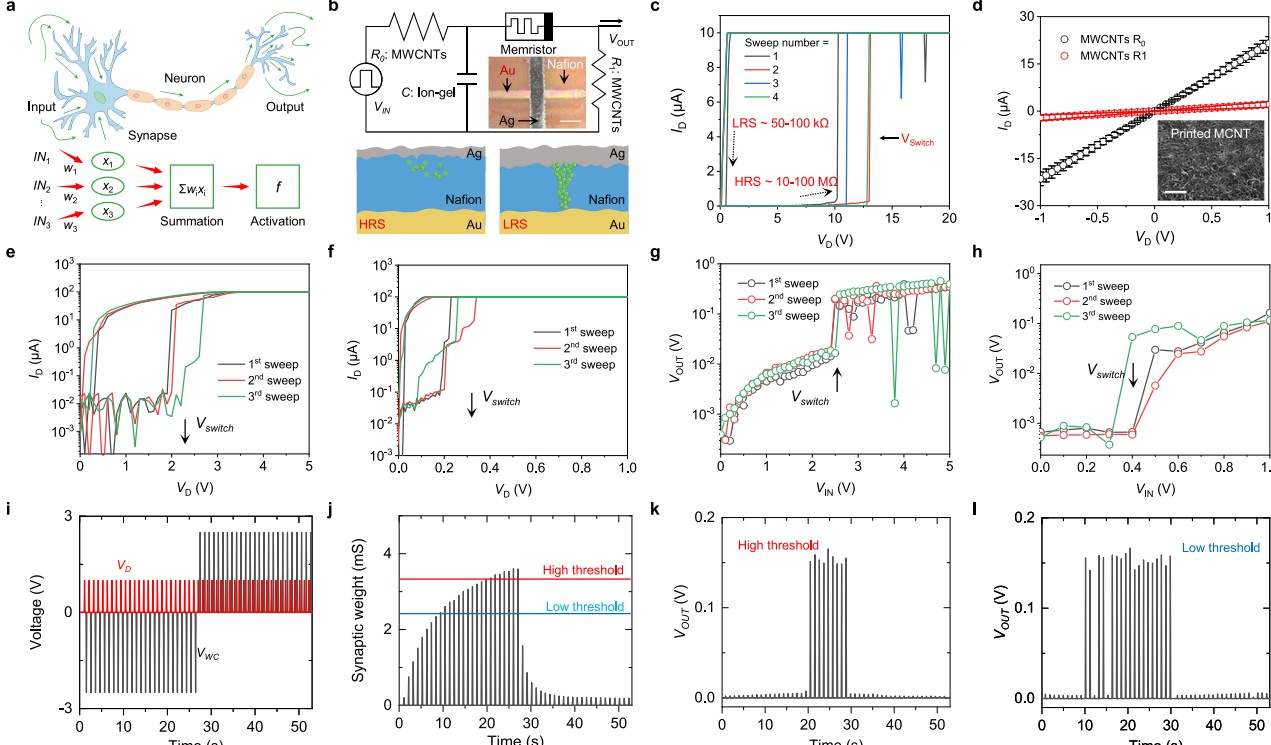

**Fig. 5 | Design and characterization of wearable synaptic nodes. a** Schematic illustration of analog data processing through a synaptic node, showing threshold-based decision-making. **b** Circuit diagram and operational mechanism of the inkjet-printed synaptic node circuit. Inset, optical image of the printed synaptic node. Scale bar, 200 μm. **c** Repeated switching curves of the printed Nafion-based memristor. **d** Current-voltage (I–V) characteristics of the inkjet-printed resistors. Inset, SEM image of a printed MWCNT register. Error bars represent s.d. of the mean from 5 different samples. **e,f** Switching curves of the memristor with different threshold voltages. Memristor switching voltages were set to 2 V (**e**) and 0.3 V (**f**), respectively, by adjusting Nafion thickness and Ag line width. **g, h** Input vs. output voltage plots of the node circuits incorporating memristors with switching voltages of 2 V (**g**) and 0.3 V (**h**). **i** Configuration of input pulse signals applied to the weight-control terminal (black) and read terminal (red) of the synapse-node integrated device. **j** Real-time conductance change in the synaptic channel and threshold levels of two node circuits with distinct threshold settings. **k, l** Real-time node output recorded from a high-threshold node (**k**) and a low-threshold node (**l**).

high-set and low-set nodes are shown in Fig. 5k, l. When the synapse conductance was below the threshold, the node output remained negligible at sub-mV levels. However, once the threshold was exceeded, the output generated a distinct output pulse of 0.15 V.

Importantly, the synapse-node integrated circuit demonstrated excellent mechanical flexibility and electrical durability, ensuring reliable performance in wearable environments (Supplementary Figs. 31 and 32). Our synaptic nodes function as threshold-based processors, offering greater adaptability and suitability for wearable devices compared to conventional complementary metal-oxide semiconductor (CMOS) circuits. This design enables for the seamless integration of diagnostic algorithms, representing a key advancement in developing hardware neural networks capable of analog signal processing for wearable neuromorphic computing applications.

## Wearable neuromorphic system for sepsis diagnosis and monitoring

The development of CSPINS integrates advancements in sensor technology, processing units, and neural networks outlined in the preceding sections, resulting in a compact and wearable system tailored for medical applications such as sepsis diagnosis and monitoring. Sepsis, a life-threatening condition caused by bacterial infections, is classified based on symptoms including fever, elevated HR, and the bacteria infection in the bloodstream (reflected by elevated lactate levels) (Fig. 6a)[43,44]. Prompt diagnosis and continuous monitoring are critical for effective management and differentiation from other conditions with similar symptoms. To demonstrate its clinical utility,

CSPINS has been engineered for sepsis diagnosis and monitoring, enabling multiplexed sensing of lactate, CBT, and HR, and processing this multimodal data through synaptic circuits and neuron-like decision units for real-time clinical decision-making.

To validate the use of CSPINS for sepsis diagnosis, we obtained data from 10 human subjects involving healthy participants and patients[45–47] with diagnosed systemic inflammatory response syndrome (SIRS), sepsis, and septic shock (Fig. 6b−d and Supplementary Table 3). Clinically, sepsis is typically diagnosed when SIRS, defined as a fever of 37.5 °C or higher and a HR of 90 BPM or greater, results from a microbial infection (reflected by lactate level higher than 2 mM)[43,44]. In addition, a lactate concentration higher than 4 mM is used as a prognostic diagnostic factor for septic shock[43,44]. Using this information, we implemented a simplified medical algorithm for sepsis diagnosis based on HR, CBT, and lactate concentration (Fig. 6e). This algorithm is implemented using a hardware neural network consisting of four synapses and five synaptic nodes. All sensors, amplifiers, and neuromorphic processing circuits are printed and integrated into a single wearable CSPINS device (Supplementary Figs. 33 and 34). The biocompatibility of the CSPINS components was further confirmed through a cell-growing cytotoxicity test (Supplementary Fig. 35).

For validation, biomarker data (HR, CBT, and lactate) from the healthy and patient participants were simulated as voltage signals and input into the synaptic circuits of CSPINS. The synaptic currents and thresholds of nodes were defined based on validated data (threshold ΔPSC: nodes 1, 2, and 5−17 μA; node 4−10 μA). The pulse signal required to operate both the sensor, and the processing circuits share the same

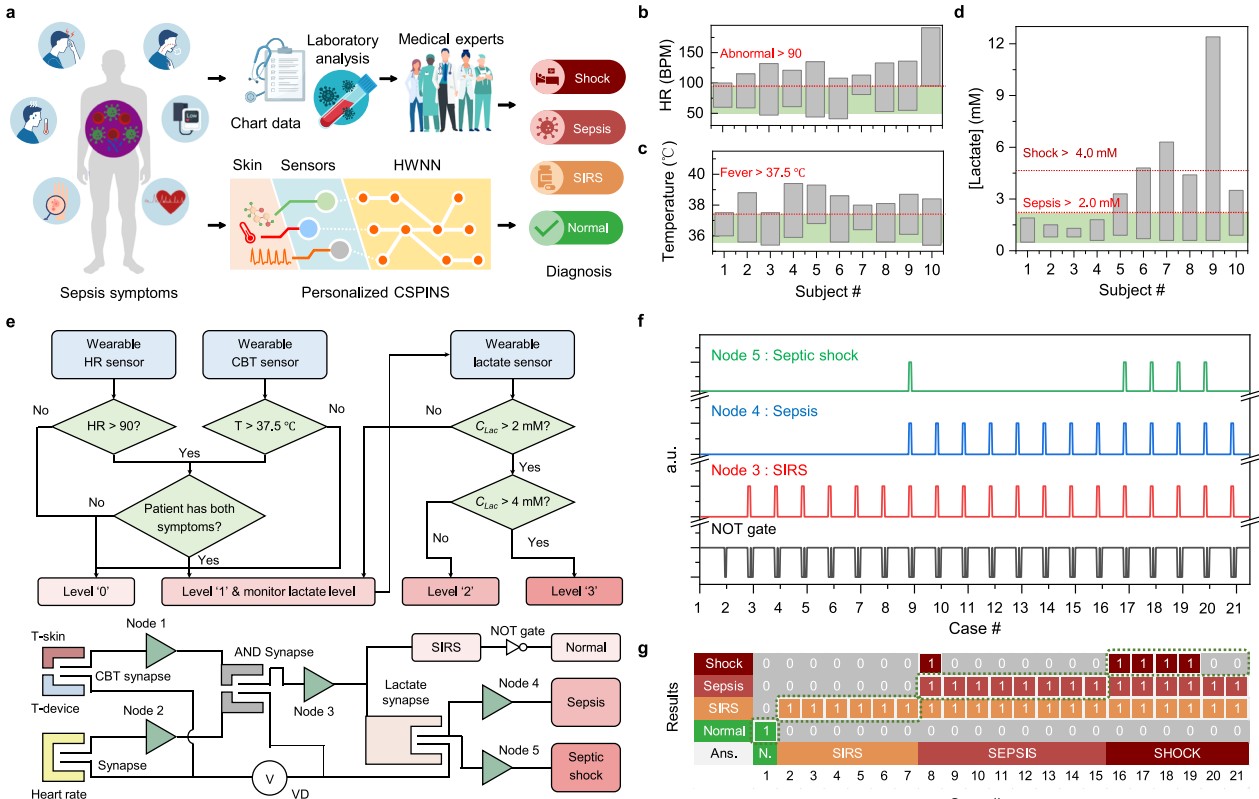

**Fig. 6 | Evaluation of the CSPINS for sepsis diagnosis and monitoring.**
**a** Schematic of the medical process for diagnosing sepsis and the operation of the wearable neuromorphic device. SIRS, systemic inflammatory response syndrome; HWNN, hardware neural network. **b**–**d** Data distribution plots of key biomarkers from healthy and patient participants: HR (**b**), CBT (**c**), and lactate (**d**). **e** Simplified sepsis diagnosis algorithm implemented in CSPINS, supported by a synapse-node network circuit design for real-time decision-making. **f** Signal outputs from four terminals in response to input data from healthy participants and patient cases, showing distinct responses for different conditions. **g** Evaluation results of the wearable CSPINS system after validation test, demonstrating its diagnostic performance for sepsis classification.

frequency (2 Hz), ensuring that the synaptic currents remain synchronized within 1–20 µA level. The PSC values and output signals of each synapse and node are demonstrated in Fig. 6f, g, and Supplementary Table 4. The neural network's decision-making process relied on nodes 3 through 5, which were responsible for identifying SIRS, sepsis, and septic shock, respectively. Healthy cases showed no output activation, SIRS cases activated node 3, sepsis cases triggered nodes 3 and 4, and septic shock cases activated all three synaptic nodes. The CSPINS demonstrated accurate diagnostic capabilities (Fig. 6g), leveraging its hardware neural network to efficiently process large volumes of biomarker data in real time. In addition, after incorporating 11 common non-inflammatory cases, such as hypertension and diabetes, we conducted a diagnostic simulation (Supplementary Tables 5 and 6). The results confirmed an overall accuracy of 84.4%, as evidenced by the confusion matrix (Supplementary Fig. 36). Finally, the estimated power consumption of CSPINS was comprehensively evaluated in Supplementary Fig. 37 and Supplementary Tables 7 and 8. The power required for processing a single case was 37.6 µW for synaptic operations and 5.3 mW when considering the entire system, demonstrating higher power efficiency compared to conventional medical diagnostic systems[4,48]. Notably, even a small communication module in commercial wearable devices requires power in the mW range, further emphasizing the energy efficiency of CSPINS. These findings reinforce the feasibility of CSPINS for continuous, low-power health monitoring. It is important to note that while our current system relies on fixed threshold values, future iterations could incorporate adaptive threshold tuning via electrical control, enabling dynamic response adjustments over time. In addition, integrating motion recognition and temporal information processing could further enhance diagnostic accuracy, ensuring robust and reliable classification even in real-world, ambulatory settings.

In this work, we have developed a fully integrated, chip-less wearable neuromorphic system that combines advanced sensor technologies, analog processors, and hardware neural networks to enable real-time biomedical signal processing and clinical decision-making. By leveraging scalable inkjet printing fabrication techniques, we designed and manufactured flexible and skin-conformal devices capable of continuous multimodal sensing and on-device computing. CSPINS demonstrates the ability to collect and process diverse multi-modal physiological data—including molecular biomarker levels, CBT, and HR—through a network of artificial synapses and neuron-inspired circuits, providing a robust platform for wearable healthcare applications. Furthermore, the integration of synaptic processors within wearable systems offers new possibilities for real-time, energy-efficient, and secure biomedical computing. We successfully validated CSPINS for sepsis diagnosis and monitoring, showcasing its capability to integrate diverse biomarkers into a simplified medical algorithm implemented via a neuromorphic processor. By overcoming key challenges such as mechanical flexibility, signal variability, and scalability limitations, CSPINS offers a transformative solution for real-time, accessible, and low-cost diagnostic tools. This scalable, low-cost approach not only enhances usability but also opens pathways for addressing other complex medical conditions, positioning CSPINS as a versatile platform for advancing wearable healthcare technologies.

## Methods
### Materials
To fabricate the electrodes of each sensor and processor circuit, commercial gold ink (Drycure Au-J 0410B) and silver ink (Metalon® JS-A102A) were purchased from C-INK Co. Poly(3-hexylthiophene-2,5-diyl) regioregular (P3HT), Nafion solution, chitosan, poly(ethylene

glycol) diacrylate (PEGDA), 2-hydroxy-2-methylpropiophenone (HOMPP), indium nitride hydrate and poly(4-vinyl phenol) (PVP), poly(melamine-co-formaldehyde) (PMF), and propylene glycol methyl ether acetate (PGMEA) were purchased from Sigma Aldrich Inc. Semiconductive single-walled carbon nanotubes (SWCNTs) and metallic multi-walled carbon nanotube (MWCNTs) were sourced from NanoIntegris co. 1-2 dichlorobenzene, 1-cyclohexyl-2-pyrrolidinone (CHP), SU-8 2000 solution, 2-methoxyethanol (2MeOH) and bis(trifluoromethylsulfonyl)azanide;1-ethyl-3-methylimidazol-3-ium [EMIM] [TFSI] ionic liquid (IL) were purchased from Fisher Scientific Co. Flexible substrates including polyimide (PI) and polyethylene terephthalate (PET) films were purchased from Kapton®. To print our electronic devices, an inkjet printer of Dimatix Materials Printer DMP-2850 equipped with SAMBA cartridges was employed.

### Preparation of artificial synapse
Before printing the Au electrodes, the PI substrate was cleaned with oxygen plasma for 15 min. The commercially available Au ink was printed precisely at a plate temperature of 45 °C. Next, SWCNT ink (0.05 mg ml$^{-1}$ concentration in CHP) was patterned, and the substrate was dried at 180 °C for 30 min. P3HT ink with 7 mg ml$^{-1}$ concentration in 1-2 dichlorobenzene was subsequently printed and dried under a nitrogen-purged acryl box for 12 h at room temperature. Residual SWCNTs were removed using 15 min of oxygen plasma treatment. Finally, the ion-gel ink (ratio of IL:PEGDA:HOMPP = 22:2:1 and added 10 w% of isopropyl alcohol to control the viscosity) was printed at 50 °C and cured under ultra-violet light exposure for 10 s.

### Fabrication and characterizations of synaptic biochemical sensors
An electrochemical workstation (CHI 760E, CH Instruments) was used to functionalize the gate electrode of the P3HT synaptic transistors and the In$_2$O$_3$ transistors. Prussian blue nanoparticles (PBNPs) were deposited onto the gold gate electrode of P3HT synaptic transistors by applying potential steps cycles (− 0.2 V versus Ag/AgCl for 1 s, 0.3 V for 1 s, 10 cycles) in a solution containing 2.5 mM FeCl$_3$, 2.5 mM K$_3$Fe(CN)$_6$, 100 mM KCl and 100 mM HCl. Pt nanoparticles (PtNPs) were deposited onto the gold gate electrode of In$_2$O$_3$ transistors through chemical reduction in a solution of 2.5 mM H$_2$PtCl$_6$ and 1.5 mM formic acid (−0.1 V versus Ag/AgCl for 700 s) to form the PtNPs-coated gold gate electrodes. Before enzyme deposition, the electrodes were dried, and 2 μl of glucose or lactate enzyme cocktail (40 mg enzyme dissolved in 1 ml PBS) was drop-cast onto each functionalized gate electrode surface. The electrodes were dried overnight at 4 °C. The Ag/AgCl reference electrode was prepared by drop-casting 0.5 μl of 0.1 M FeCl$_3$ onto the silver surface for 40 s, followed by rinsing with deionized water.

### Preparation of amplifiers
The amplifiers were patterned through sequentially printing of gold, SWCNT, cross-linked PVP, MWCNTs, and an additional gold layer using a DMP-2850 printer. The gold substrate was first printed to form the connections and transistor channels, followed by annealing in an oven at 180 °C for 30 min. Next, the SWCNT layer was printed and annealed on a hotplate at 180 °C for 30 min. Dielectric layers were fabricated by printing PVP ink (doped with 1.25% v/v polymer ion liquid) layer by layer. After printing each layer, the patch underwent soft baking in an oven at 180 °C for 30 s. Upon completion of all layers, the PVP dielectric was cross-linked and annealed on a hotplate at 180 °C for 40 min. Subsequently, the MWCNTs were patterned in specific regions to form the resistors. Finally, an additional gold layer was printed on top of the cPVP dielectric layer to complete the amplifier structure.

### Preparation of synaptic node circuit
A Nafion ink, composed of Nafion, ethanol, and ethylene glycol in a 1:1:2 ratio, was printed onto pre-patterned Au electrodes. A total of 12 layers of Nafion ink were printed in three cycles, with the water content of the memristive channels controlled by drying at 120 °C for 10 min between each printing cycle. For samples requiring interconnections with other circuits, SU-8 insulating layers were printed at the anticipated electrode intersection points to prevent short circuits. The samples were then dried at 180 °C for 30 min.

Subsequently, the dried samples underwent oxygen plasma treatment for 15 min to equalize surface energy. MWCNT ink was then printed under optimized conditions to achieve the desired resistance level. Finally, silver patterns were printed onto the dried Nafion layer, and the entire sample was dried again at 180 °C for 30 min to complete the fabrication process.

### Cytocompatibility test of CSPINS
Normal Adult human dermal fibroblasts (HDFs) cells (Lonza) were cultured under 37 °C and 5% CO$_2$ and subcultured at 70% confluence. Cells at passages 4 to 6 were used. The CSPINS patch was washed with 70% ethanol and transferred to 24-well cell culture inserts. HDFs were seeded at a density of $1 \times 10^5$ cells per well, and the inserts were then placed in cell-seeded wells. The cells were then treated with fibroblast basal medium supplemented with fibroblast growth kit components (ATCC) and incubated at 37 °C and under 5% CO$_2$ during the study. Inserts without CSPINS served as controls. Cell viability and metabolic activity were evaluated using a LIVE/DEAD™ Viability/Cytotoxicity Kit (Invitrogen) and PrestoBlue assays (Thermo Fisher Scientific), respectively. For the live/dead assay, live cells were stained green (calcein-AM) and dead cells red (ethidium homodimer-1) and imaged with a ZEISS Axio Observer inverted microscope. In the PrestoBlue assay, 500 μl of medium with 10% v/v PrestoBlue reagent was added per well, incubated for 45 min at 37 °C, and transferred to a 96-well plate for fluorescence measurement (ex 540 nm/em 590 nm) using a BioTek plate reader.

### Evaluation of the CSPINS in human subjects
The evaluation of CSPINS was conducted in human subjects strictly adheres to established ethical guidelines as delineated in protocols approved by the Institutional Review Board (IRB) at the California Institute of Technology (Caltech) (#IR22-1280). The participants were recruited from both the Caltech campus and nearby communities around Los Angeles, California. All study participants provided written informed consent prior to their involvement in the research.

### Reporting summary
Further information on research design is available in the Nature Portfolio Reporting Summary linked to this article.

## Data availability
The data that support the findings of this study are available from the corresponding author upon request.

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

## Acknowledgements

This project was supported by the National Science Foundation grant 2145802 and 2444815, National Institutes of Health grants R01HL155815 and R01DC021461, Army Research Office grant W911NF-23-1-0041, US Army Medical Research Acquisition Activity grant HT9425-24-1-0249, and Heritage Medical Research Institute. Y.C. acknowledged the support from the Basic Science Research Program funded by the Ministry of Education (2021R1A6A3A03046099), Korean Institute for Advancement of Technology (KIAT) grant funded by the Korea Government (RS-2024-00435693). S.L. acknowledged the support from the BrainLink program funded by the Ministry of Science and ICT through the National Research Foundation of Korea (RS-2023-00237308 and 2020R1A5A1018052).

## Author contributions

Y.C. and W.G. conceived the idea. Y.C., P.J. and S.L. led the research efforts and performed the main experiments. Y.S., R.Y.T., G.K., J.Y., H.H. and J.Y. assisted in the experiments and analysis of results. W.G., J.H.C. and D.K. supervised the studies. Y.C. and W.G. wrote the paper. All

authors contributed to the data analysis and provided feedback on the manuscript.

## Competing interests

The authors declare no competing interests.
