## [Transparent Peer Review file · Nature Communications]

All-printed chip-less wearable neuromorphic system for multimodal physicochemical health monitoring

Corresponding Author: Professor Wei Gao

Version 0:

Reviewer comments:

Reviewer #1

(Remarks to the Author)

The manuscript by Choi et al. describes an all-printed neuromorphic circuit that utilises and senses biosignals, encodes these into spikes which are then processed through the circuit and used to classify sepsis. The manuscript is very well written and was a pleasure to read. Figures are clear and informative. The results are impressive and although the synapses based on slow ion kinetics have been demonstrated before, the combination of multiple input signals, processing, and classification on single printable chip, is very impressive. I recommend publication of this paper after a few minor comments can be addressed:

1. Some parts of the system utilises pulses from the signal it measures itself, like the heartbeat while other pulses are generated (because they originate from a static source like temperature). How are these pulses matched, if at all, and does that mean that for this wearable system an (external) pulse generator is still necessary?
2. The authors mention supervised training (Fig 6e). What part of the system is trained? Are the weights trained in software and programmed afterwards?
3. Related to that, Figure 4h mentions "weight-update". What do the authors mean here exactly? Are there analog weights that are trained/programmed?

Reviewer #2

(Remarks to the Author)

The authors of this work developed a standalone neuromorphic system capable of monitoring, processing, and analysing different biorelevant signals. Specifically, they integrated in their platform (CSPINS) a glucose and lactate biosensors, a temperature sensor and a heart rate sensor. They combined different devices based on both organic and conventional electronics technologies. The signals coming from these sensors were directly transmitted to a synaptic transistor based on P3HT-SWCNT to "encode" them into memory states, replicating the functions of a biological synapse. These states were then summed on a vertical crossbar (Au/Nafion/Ag) memristor to allegedly emulate the functions of a biological neuron. Finally, the authors implemented a simple medical diagnostic algorithm for processing and decision making of the data processed by the memristor.

Due to the presented analysis of the neuromorphic performance of the system, the combination of employed techniques and established technologies, and more importantly to the inaccurate discussion of the neuromorphic foundations of this platform's design, I believe that at the current stage the manuscript is not appropriate for publication in Nature Communications.

In particular, the reviewer raises some general concerns on the various neuromorphic mechanisms presented besides the claim of their not-incremental significance, specifically regarding the use of the word "neuron" and its significance not only in the context of this work but more generally in the neuromorphic engineering field.

Hence, the reviewer reports here the major aspects to be addressed in order to aid the manuscript for later submission:

(1) The reviewer asks to present a complete analysis of the synaptic device system, with a dedicated section in the supporting information and additional information in the main text. Particularly, it is difficult to understand which are the frequency limits of the presented system, i.e. which is the capability of the synaptic device to store the information coming from a fast input signal. Also, the reviewer asks for a complete evaluation of the stability over time of this platform, under operation and different interfacing scenarios.

(2) Referring to Fig.4, which is the neuromorphic response of the platform for heart rates below 80BPM, would this signal be distinguishable from the sensing window provided in the text?

(3) As mentioned above, Figure 5 depicts a “neuron” circuit and its output (used for the medical algorithm). Neurons have reproducible uniform outputs which are event-based and constituted by rapid burst of action potentials (spikes). The intrinsic biological definition of neurons is inconsistent with the data presented (particularly Fig 5 l), where the memristor responds to a pulsed input with a pulsed output. This mechanism does not replicate the spike-encoding of a biological system, rather working closely to a synaptic pre-processing. Moreover, in Fig.5j,k,l the high and low threshold are poorly defined, as the low threshold shows a variability close to a background signal rather than a stable output signal. The authors are here invited to comment and clarify on the “neuron” device operative functions and to consider the use of an alternative definition for the functions of this device (synaptic modulator, synaptic node).

(4) The reviewer asks to precisely comment on the voltages increase (spikes) in Fig 5h with large, variable amplitude, considering that conventional memristor circuits of similar design display uniform outputs which can be more easily implemented with algorithms. If there is a materials or device explanation, please provide a justification for the choice of the adoption of this particular device.

(5) Please provide in the main text a throughout description of the benefits and potential application of such flexible system.

(6) The authors should provide a full evaluation of the power consumption of CSPINS. On this basis, they should provide and demonstrate the use of wearable or conformable power unit and validate the functions of the whole system. Additionally, they should benchmark the performance of the presented system respect to the metrics of a biological circuit performing similar functions. Which is the biology-to-neuromorphic power consumption gap? How can this be overcome with future design/materials combinations?

(7) In Figure 6e, a clear flow chart is presented with and without von-Neumann architectures, to explain the function of the artificial “neurons” in parallel with the digital processing represented by AND and NOT gates.

As such, what is the benefit of utilizing “neurons” in this case as a clear system can be defined (based of binary logic) and the “neurons” serve no function other than to integrate inputs? If the “neurons” are simply acting as threshold devices could this algorithm be implemented using another (simpler) circuitry? If this is to improve power consumption, what is the power consumption of the system, and would it not be lower removing either the von-Neumann component or the neurons to enable more efficient architectures? A key advantage of neuromorphic systems is the ability of the system to modulate/learn over time and on an event-driven basis.

As such, are the thresholds (red and blue in Fig.5) determined by circuit design? This would limit the circuit leading to ineffective “training”, in which case, what training is done? For example, if the threshold are fixed by materials/circuit design, what is the point of this circuit since for example the level corresponding to “exercise” would be classified at level 1 with SIRS regardless of sepsis risk, since there is no motion information or even temporal information about the sensors?

(8) Figure 6G: What is the accuracy presented here? Please include a confusion matrix, as well as enough normal samples for significance.

Version 1:

Reviewer comments:

Reviewer #1

(Remarks to the Author)

The authors have successfully addressed my comments. The manuscript can now be accepted.

Reviewer #3

(Remarks to the Author)

The reviewer acknowledges that the other comments have been well addressed; however, there is still an issue regarding power consumption. The authors claimed that the system demonstrates advantages in terms of power consumption. However, throughout the manuscript, there is no supporting data comparing the power or energy consumption of this system with that of conventional CMOS-based systems. Please include a comparison table for power consumption in the manuscript, and provide a detailed explanation of the calculation processes.

REVIEWER COMMENTS

We thank both reviewers for carefully reading our manuscript and providing highly valuable suggestions. We have revised the manuscript as per the reviewers' suggestions. The revised text is highlighted in red in the main text and below is a response to each comment.

Reviewer #1

The manuscript by Choi et al. describes an all-printed neuromorphic circuit that utilises and senses biosignals, encodes these into spikes which are then processed through the circuit and used to classify sepsis. The manuscript is very well written and was a pleasure to read. Figures are clear and informative. The results are impressive and although the synapses based on slow ion kinetics have been demonstrated before, the combination of multiple input signals, processing, and classification on single printable chip, is very impressive. I recommend publication of this paper after a few minor comments can be addressed

Reply: We thank the reviewer for carefully reading our paper and providing us with insightful and constructive feedback to further improve our paper. We have carefully revised the manuscript by adding discussions to fully address all the concerns raised by the reviewer. We believe that these changes have further improved our paper.

Comment 1. Some parts of the system utilises pulses from the signal it measures itself, like the heartbeat while other pulses are generated (because they originate from a static source like temperature). How are these pulses matched, if at all, and does that mean that for this wearable system an (external) pulse generator is still necessary?

Reply: Thank you for your valuable comments. We appreciate your thoughtful feedback and would like to clarify the detailed operation of our system.

In our proposed system, all processing units—except for the heart rate (HR) acquisition unit—operate based on pulse signals, similar to other artificial synapse-based studies. While HR data can be processed directly using a DC voltage, a pulse generator and an H-bridge circuit are required for core body temperature (CBT) calculation.

To match the pulse form, the synaptic current generated by the HR signal and CBT processing are synchronized first. By setting the weight-update time of 3 seconds for the HR sensor and inputting 6 pulses for CBT calculation, the resulting synaptic currents for each unit were matched at the 1–20 μA level. Then, the current information is processed through the synapse-neuron connections. The synaptic current above the neuron's threshold results in a pulse signal when we input the pulse to operate the synapse-neuron circuit. At this stage, the pulse pattern aligns with that used to operate the synaptic biochemical sensor, ensuring consistency in subsequent calculations.

Since the pulse signal required to operate the sensor units and operating synapse-neuron circuit share the same frequency, a single pulse generator is sufficient, eliminating the need for additional synchronization process. Therefore, at this stage, we believe that a simple circuit unit incorporating a DC unit and a pulse generator is sufficient for implementing our CSPINs device.

We recognize that synchronization between signals is a key point of emphasis in this study. To clarify this, we have made the following revision in the main text.

In page 10: *“The synaptic currents and thresholds of nodes were defined based on validated data (threshold ΔPSC : neurons 1, 2, and 5—17 μA ; neuron 4—10 μA). The pulse signal required to operate both the sensor and the processing circuits share the same frequency (2 Hz), ensuring that the synaptic currents remain synchronized within 1–20 μA level.”*

Comment 2. The authors mention supervised training (Fig 6e). What part of the system is trained? Are the weights trained in software and programmed afterwards?

Reply: Thank you for your insightful question. The system undergoes supervised testing based on 20 known case data samples. While this testing process shares similarities with an artificial intelligence training procedure, it does not involve traditional machine learning training. In our system, the weights are not trained in software and then programmed afterward. Instead, the weights are determined based on predefined parameters from the case data and applied directly during the testing phase.

We recognize that using the term "supervised training" may have caused some confusion. To address this, we revised the section to provide a clearer explanation of the procedure, distinguishing it from traditional machine learning training methods.

In page 24: *“g, Evaluation results of the wearable CSPINS system after validation test, demonstrating its diagnostic performance for sepsis classification.”*

Comment 3. Related to that, Figure 4h mentions “weight-update”. What do the authors mean here exactly? Are there analog weights that are trained/programmed?

Reply: Thank you for your valuable comments. In the manuscript, we initially used the term "weight-update" to describe the process of inputting physiological data and converting it into synaptic current. However, we recognize that this term may cause confusion, as it is typically associated with training or adjusting weights in machine learning systems. In our case, there are no analog weights that are trained or programmed. Instead, physiological signals (such as heart rate, lactate, etc.) are input into the system and directly converted into synaptic currents.

To clarify, our system does not involve trained or programmed analog weights. Instead, physiological signals (such as heart rate, lactate, etc.) are directly input into the system and converted into synaptic currents. To improve clarity, we replaced the term "weight-update" with more precise terminology, such as "biosignal input" or "pulse input," to better reflect the process of encoding biosignals into synaptic currents.

In page 6: *“CBT mining was executed by connecting the T_{skin} and T_{AI} nodes to a multi-gate synapse, which implemented the calculation through sequential biosignals input (Fig. 3j). The process involved three initial pulse input to encode the term $(T_{skin} - T_{AI}) \times R_{skin}/R_{device}$, followed by three additional updates to add T_{skin} .”*

In page 8: *“As HR increased, more pulse signals were delivered to the synapse during the biosignal input window, resulting in higher PSC values (Fig. 4i).”*

In page 21: “*i, Additive and subtractive characteristics of the synaptic calculator with two biosignals input terminals. j, Real-time biosignal input plots obtained from attaching CBT sensors to various body parts in different environments.*”

Reviewer #2

General Comment. The authors of this work developed a standalone neuromorphic system capable of monitoring, processing, and analysing different biorelevant signals. Specifically, they integrated in their platform (CSPINS) a glucose and lactate biosensors, a temperature sensor and a heart rate sensor. They combined different devices based on both organic and conventional electronics technologies. The signals coming from these sensors were directly transmitted to a synaptic transistor based on P3HT-SWCNT to “encode” them into memory states, replicating the functions of a biological synapse. These states were then summed on a vertical crossbar (Au/Nafion/Ag) memristor to allegedly emulate the functions of a biological neuron. Finally, the authors implemented a simple medical diagnostic algorithm for processing and decision making of the data processed by the memristor.

Due to the presented analysis of the neuromorphic performance of the system, the combination of employed techniques and established technologies, and more importantly to the inaccurate discussion of the neuromorphic foundations of this platform’s design, I believe that at the current stage the manuscript is not appropriate for publication in Nature Communications.

In particular, the reviewer raises some general concerns on the various neuromorphic mechanisms presented besides the claim of their not-incremental significance, specifically regarding the use of the word “neuron” and its significance not only in the context of this work but more generally in the neuromorphic engineering field.

Hence, the reviewer reports here the major aspects to be addressed in order to aid the manuscript for later submission.

Reply: We thank the reviewer for carefully reading our paper and providing us with insightful and constructive feedback to further improve our paper. We have carefully revised the manuscript by adding discussions to fully address all the concerns raised by the reviewer. We believe that these changes have further improved our paper.

Comment 1. The reviewer asks to present a complete analysis of the synaptic device system, with a dedicated section in the supporting information and additional information in the main text. Particularly, it is difficult to understand which are the frequency limits of the presented system, i.e. which is the capability of the synaptic device to store the information coming from a fast input signal. Also, the reviewer asks for a complete evaluation of the stability over time of this platform, under operation and different interfacing scenarios.

Reply: Thank you for your valuable feedback. To address the reviewer's request for a complete analysis of the synaptic device system, we have conducted measurements at various frequencies to evaluate the frequency limits of the system (**Fig. R1**). Our device operates effectively up to approximately 100 Hz, which is sufficient for processing typical biosignals such as heart rate (typically ranging from 0.5 to 2 Hz). While it does not operate at extremely high frequencies, this range is adequate for the intended applications in wearable biosignal processing. The relatively low operating frequency is primarily due to the large dimension, as

it is fabricated cost-effectively via inkjet printing. This results in slower response times compared to smaller cleanroom-fabricated devices. However, we believe that performance can be improved through future optimization efforts, such as enhancing material properties or further miniaturizing the device, which may increase speed without compromising functionality.

Regarding stability and long-term performance, we subjected the synaptic device to rigorous testing under challenging conditions. The device demonstrated excellent stability, maintaining current characteristics even after 4000 repetitive bending tests (**Fig. R2**) and 3000 cycles of write/erase operations (**Fig. R3**). These results highlight the durability and reliability of the platform, even under mechanical stress and prolonged use.

Fig. R1 | Frequency-dependent characterization of synaptic performance. a–c, Long-term potentiation/depression characteristics of the synapse under varying input frequencies: 10 Hz (a), 100 Hz (b), and 1 kHz (c).

Fig. R2 | Bending durability evaluations of the synapse. a, Real-time changes in post-synaptic current (PSC) under repeated bending cycles. b, Evolution of excitatory post-synaptic current (EPSC), inhibitory post-synaptic current (IPSC), and maximum conductance (G_{max}) during repeated bending tests.

Fig. R3 | Electrical durability evaluations of the synapse. **a**, Real-time PSC changes of the synapse during repeated write/erase operations over 3000 cycles. **b–d**, Detailed plot of PSC changes at 0 s (**b**), 3000 s (**c**), and 6000 s (**d**) after measurement.

We have included the corresponding data in the revised main text and supporting information to provide a comprehensive evaluation of the system's stability as follows:

In page 9: “Importantly, the synapse-node integrated circuit demonstrated excellent mechanical flexibility and electrical durability, confirming reliable performance in wearable environments (**Supplementary Figs. 31 and 32**).”

In Supplementary Information:

Supplementary Fig. 31 | Bending flexibility evaluations of the wearable neuromorphic system. *a*, Optical images of the wearable neuromorphic system under different bending angles. *b*, Characterization of synaptic currents of artificial synapse unit under different bending angles. *c*, Characterization of Nafion memristors under different bending angles. *d*, Characterization of synapse-neuron integrated circuit under different bending angles. Th , threshold. *e*, Real-time changes in PSC under repeated bending cycles. *f*, Evolution of EPSC, IPSC, and maximum conductance (G_{max}) during repeated bending tests.

Supplementary Fig. 32 | Electrical durability evaluations of the synapse. *a*, Real-time PSC changes of the synapse during repeated write/erase operations over 3000 cycles. *b–d*, Detailed plot of PSC changes at 0 s (*b*), 3000 s (*c*), and 6000 s (*d*) after measurement.

Comment 2. Referring to Fig.4, which is the neuromorphic response of the platform for heart rates below 80 BPM, would this signal be distinguishable from the sensing window provided in the text?

Reply: Thank you for your insightful comment. To address the reviewer's concern regarding the neuromorphic response for heart rates below 80 BPM, we conducted additional experiments to assess the system's performance within a wider range. We confirmed that the typical heart rate range for healthy adult males starts from around 60 BPM.

To provide a more comprehensive evaluation, we performed additional measurements for heart rates between 60 and 70 BPM (**Fig. R4**). The results follow the trend observed in the existing data and exhibit a linear decrease, supporting the system's ability to accurately process physiological signals within this range. Based on these extended measurements, we believe that the neuromorphic platform can effectively distinguish signals within this heart rate range. We have included this additional data and updated the corresponding plot in the revised manuscript to enhance clarity.

Fig. R4 | Characterization of the wearable heart rate processor at lower BPM. a,b, Sensor response under heart rates around 60 BPM (a), and 70 BPM (b). c, Real-time PSC changes updated under relaxed body conditions. d, Calibration curve comparing PSC changes from the synaptic HR processor with readings from the commercial HR monitor.

We have now added the new data in revised **Fig. 4**:

Fig. 4 | Design and characterization of the wearable processor for heart rate (HR) monitoring. a, Schematic illustration of the wearable HR monitoring system, which translates cardiovascular signals into synaptic currents. b, Schematics of the HR processing module components, comprising the receptor (pressure-sensitive sensor) and processor (synaptic

device). **c**, Optical images of a wearable HR processing module, showing the integration of skin-attached receptors and processors. Scale bar, 1 cm. **d**, Calibration plot of the conductive sponge-based pressure sensor under varying pressure levels. I and I_0 represent the current responses under applied pressure load and baseline (no pressure load), respectively. Error bars represent s.d. of the mean from 5 different samples. $V_D = 1$ V, $I_0 \sim 26$ μ A. **e**, Resistance variations of the conductive sponge under different body conditions. R and R_0 represent the resistance under exercise and at rest, respectively. $R_0 \sim 38$ k Ω . BPM, beats per min. **f**, Voltage signal output after processing the resistance changes through the amplifier circuit. **g**, Calibration curve correlating the frequency of the amplified voltage pulses with a commercial HR monitor. **h**, Characterization of the PSC changes during HR module operation, with readings taken after 3 s of biosignals input, followed by a reset. **i**, Real-time PSC changes updated under varying body conditions. Inset numbers were collected using a commercial HR monitor. **j**, Calibration curve comparing PSC changes from the synaptic HR processor with readings from the commercial HR monitor. **k**, Continuous 2-hour HR monitoring with a commercial HR monitor (black) and the wearable synaptic HR processor (red) during various activities performed by the subject.

Comment 3. As mentioned above, Figure 5 depicts a “neuron” circuit and its output (used for the medical algorithm). Neurons have reproducible uniform outputs which are event-based and constituted by rapid burst of action potentials (spikes). The intrinsic biological definition of neurons is inconsistent with the data presented (particularly Fig 5 I), where the memristor responds to a pulsed input with a pulsed output. This mechanism does not replicate the spike-encoding of a biological system, rather working closely to a synaptic pre-processing. Moreover, in Fig.5j,k,l the high and low threshold are poorly defined, as the low threshold shows a variability close to a background signal rather than a stable output signal. The authors are here invited to comment and clarify on the “neuron” device operative functions and to consider the use of an alternative definition for the functions of this device (synaptic modulator, synaptic node).

Reply: We sincerely appreciate the reviewer’s insightful comments. We acknowledge that while our proposed device is inspired by the threshold-firing mechanism of biological neurons, it does not replicate spike-based encoding in the same way as biological neurons. Instead, our device primarily functions as a synaptic pre-processing unit, where it integrates input signals and produces a sharp transition in output voltage once the threshold is exceeded.

To avoid any misunderstanding, we have revised the manuscript's terminology, replacing “neuron” with the more precise term “synaptic node”. This adjustment better reflects the actual functional role of the device and aligns with its operation as a signal-processing unit within the system.

Regarding the reviewer’s concern about the high and low threshold definitions in **Fig. 5j–l**, we recognize that the previously presented low threshold exhibited variability close to the background signal, making it less distinguishable as a stable output. To address this issue, we conducted further experiments to refine the threshold definitions, ensuring a well-defined and reproducible separation between different threshold levels. The printing/drying conditions of Nafion film were optimized. The new results (**Fig. R5**) demonstrate a distinct and stable signal

response, corresponding to three separate threshold levels in the synaptic node. These findings have been incorporated into the revised manuscript for improved clarity.

Fig. R5 | Characterization of wearable synaptic node a, Real-time conductance change in the synaptic channel and threshold levels of three synaptic nodes with distinct threshold settings. **b–d**, Real-time output signal recorded from a high-threshold node (**b**), middle-threshold node (**c**) and low-threshold node (**d**).

Based on these results, we revised the manuscript and Figures. Also, we added discussion on defining the terminology ‘synaptic node’ and replaced the term ‘neuron’ with synaptic node.

In page 1: “By leveraging scalable printing technology, we fabricated artificial synapses that function as both sensors and analog processing units, integrating them alongside printed synaptic nodes into a compact wearable system embedded with a medical diagnostic algorithm for multimodal data processing and decision making.”

In page 3: “CSPINS features a neuromorphic processing layer composed of arrays of artificial synapses and nodes, forming a lightweight, skin-conformal platform that enables on-body data collection, processing, and decision making without reliance on external computing resources (**Fig. 1b**).”

Throughout the section ‘**Wearable synaptic node for neuromorphic signal processing**’:

“Analog signals inherently carry extensive information, yet to enable computation with these signals, devices capable of making binary decisions (e.g., ‘0’ and ‘1’) are essential⁴³. In biological systems, the human neural network processes analog signals accumulated in synapses through neurons based on threshold firing properties (**Fig. 5a**). Mimicking this behavior, the development of synaptic node is a critical step toward precise data processing of accumulated analog signals and enabling neuromorphic computing.

Here, we present a wearable synaptic node circuit that generates a decision signal when accumulated synaptic currents surpass a threshold, achieved via the on/off switching behavior of a memristor (**Fig. 5b**). The node circuit, including its memristor, resistors (R_0 and R_1), and capacitors, was fabricated on a thin PI film via inkjet printing (**Supplementary Fig. 26**). The memristor, a central component, features a vertical crossbar structure of Au/Nafion/Ag, leveraging Nafion for its excellent mechanical flexibility and compatibility with scalable solution-based printing processing (**Supplementary Fig. 27**). When the voltage between Au and Ag exceeds the memristor's switching voltage, Ag ions migrate through the proton-conducting Nafion layer, forming conductive filaments that transition the device from a high-resistance state (HRS) to a low-resistance state (LRS) (**Fig. 5c**). When the voltage decreases, the filaments dissolve, returning the memristor to HRS.

To ensure optimal charging and discharging behavior of the node circuit, the resistances of inkjet-printed MWCNTs resistors R_0 and R_1 were tuned to 30 k Ω and 200 k Ω , respectively (**Fig. 5d**). The memristor's switching voltage was fine-tuned between 0.1 and 16 V by optimizing the Nafion thickness and Ag line width (**Fig. 5e,f** and **Supplementary Figs. 28, 29**). The circuit's performance was evaluated by applying 10 Hz input pulses with increasing amplitude (0–20 V) and measuring the output signal (**Fig. 5g,h**). As the capacitor accumulated charge, the voltage increased until the memristor's switching threshold was reached, at which point the memristor transitions to LRS, causing a rapid voltage to increase at the output terminal (**Supplementary Fig. 30**). The threshold voltage of synaptic node closely matched the memristor's switching voltage, showing a voltage increase exceeding $10\times$ upon activation.

To demonstrate analog computing capabilities, we integrated the synaptic device with node circuits set to have distinct threshold levels. A set of read voltages (1 V) and potentiation/depression voltage pulses (± 2.5 V) were applied to the synapse's gate electrode (**Fig. 5i**). The synapse conductance varied between 0.2 to 27.1 mS, with the threshold levels of low-set node and high-set node estimated at 2.3 and 3.1 mS, respectively (**Fig. 5j**). The corresponding output signals of the high-set and low-set nodes are shown in **Fig. 5k,l**. When the synapse conductance was below the threshold, the node output remained negligible at sub-mV level. However, once the threshold was exceeded, the output generated a distinct output pulse of 0.15 V.”

Fig. R5b,d has now been included as the new **Fig. 5k,l**:

Fig. 5 | Design and characterization of wearable synaptic nodes. *a*, Schematic illustration of analog data processing through a synaptic node, showing threshold-based decision-making. *b*, Circuit diagram and operational mechanism of the inkjet-printed synaptic node circuit. Inset, optical image of the printed synaptic node. Scale bar, 200 μm . *c*, Repeated switching curves of the printed Nafion-based memristor. *d*, Current-voltage (I - V) characteristics of the inkjet-printed resistors. Inset, SEM image of a printed MWCNT register. *e, f*, Switching curves of the memristor with different threshold voltages. Memristor switching voltages were set to 2 V (*e*) and 0.3 V (*f*), respectively, by adjusting Nafion thickness and Ag line width. *g, h*, Input vs. output voltage plots of the node circuits incorporating memristors with switching voltages of 2 V (*g*) and 0.3 V (*h*). *i*, Configuration of input pulse signals applied to the weight-control terminal (black) and read terminal (red) of the synapse-node integrated device. *j*, Real-time conductance change in the synaptic channel and threshold levels of two node circuits with distinct threshold settings. *k, l*, Real-time node output recorded from a high-threshold node (*k*) and a low threshold node (*l*).

The labelling in **Fig. 6e** has also been updated:

Comment 4. The reviewer asks to precisely comment on the voltages increase (spikes) in Fig 5h with large, variable amplitude, considering that conventional memristor circuits of similar design display uniform outputs which can be more easily implemented with algorithms. If there is a materials or device explanation, please provide a justification for the choice of the adoption of this particular device.

Reply: We appreciate the reviewer’s concern regarding the voltage spikes observed in Fig. 5h and variability in amplitude. The primary source of this behavior is the proton conduction dynamics of the Nafion-based electrolyte layer. Since Nafion is a proton-conducting polymer, the memristive behavior is governed by the migration and accumulation of protons at the electrode interfaces. This enables low-voltage operation but also makes the device highly sensitive to water content, leading to variability in switching behavior.

Despite this variability, Nafion-based memristors offer key advantages that make them particularly promising for future wearable neuromorphic systems, including superior mechanical flexibility, biocompatibility, and compatibility with solution-based fabrication techniques such as inkjet printing. These properties make the device particularly promising for future wearable neuromorphic systems.

To further enhance the operational stability of both the memristor and synaptic node, we optimized the drying conditions during the Nafion layer deposition process. As a result, the output signals of the synaptic node have become more consistent, reducing the variability observed in previous data (**Fig R5**).

Considering the reviewer’s comments, we have revised the manuscript as follows:

i) The plot in **Fig. 5g,h** has been changed to a log scale to better highlight the switching point.

ii) A discussion on Nafion as the active material has been added to clarify our rationale for its selection.

iii) Additional details regarding the fabrication process of the Nafion-based memristor have been included for transparency.

In page 9: “The memristor, a central component, features a vertical crossbar structure of Au/Nafion/Ag, leveraging Nafion for its excellent mechanical flexibility and compatibility with scalable solution-based printing processing (**Supplementary Fig. 27**).”

In page 13: “Preparation of synaptic node circuit.

A Nafion ink, composed of Nafion, ethanol, and ethylene glycol in a 1:1:2 ratio, was printed onto pre-patterned Au electrodes. A total of 12 layers of Nafion ink were printed in three cycles, with the water content of the memristive channels controlled by drying at 120 °C for 10 minutes between each printing cycle. For samples requiring interconnections with other circuits, SU-8 insulating layers were printed at the anticipated electrode intersection points to prevent short circuits. The samples were then dried at 180 °C for 30 min.”

In revised Fig. 5:

Fig. 5 | Design and characterization of wearable synaptic nodes. *a*, Schematic illustration of analog data processing through a synaptic node, showing threshold-based decision-making. *b*, Circuit diagram and operational mechanism of the inkjet-printed synaptic node circuit. Inset, optical image of the printed synaptic node. Scale bar, 200 μm . *c*, Repeated switching curves of the printed Nafion-based memristor. *d*, Current-voltage (*I-V*) characteristics of the inkjet-printed resistors. Inset, SEM image of a printed MWCNT register. *e,f*, Switching curves of the memristor with different threshold voltages. Memristor switching voltages were set to 2 V (*e*) and 0.3 V (*f*), respectively, by adjusting Nafion thickness and Ag line width. *g,h*, Input vs. output voltage plots of the node circuits incorporating memristors with switching voltages of 2 V (*g*)

and 0.3 V (**h**). **i**, Configuration of input pulse signals applied to the weight-control terminal (black) and read terminal (red) of the synapse-node integrated device. **j**, Real-time conductance change in the synaptic channel and threshold levels of two node circuits with distinct threshold settings. **k,l**, Real-time node output recorded from a high-threshold node (**k**) and a low threshold node (**l**).

Comment 5. Please provide in the main text a throughout description of the benefits and potential application of such flexible system.

Reply: We sincerely appreciate the reviewer's constructive comments, which have been instrumental in refining the core identity of our system as an analog bio-signal processing unit. In response to the valuable feedback, we have made revisions in the main text to explicitly highlight the unique advantages and potential applications of our approach.

Unlike conventional sensor systems that rely on digital conversion and external computation, our system processes analog biosignals directly within an analog computing framework. This eliminates the need for analog-to-digital conversion (ADC), enabling more energy-efficient, real-time processing that is ideally suited for low-power applications. By leveraging neuromorphic principles, our system is capable of performing on-device, analog computation for various bio-signals, such as glucose, lactate, heart rate, and temperature, without requiring communication with external processors. The highly independent processing enhances computing efficiency as well as security.

Additionally, the use of flexible, solution-processable materials, which allow fabrication via inkjet printing, makes our system highly adaptable for wearable, low-power health monitoring applications. This enables the development of conformable and scalable biomedical devices capable of real-time biosensing.

To emphasize the broader impact of our work in bioelectronics and neuromorphic computing, we have integrated these key aspects into the revised manuscript, particularly in the discussion on potential applications.

In page 11: *“Furthermore, the integration of synaptic processors within wearable systems offers new possibilities for real-time, energy-efficient, and secure biomedical computing. We successfully validated CSPINS for sepsis diagnosis and monitoring, showcasing its capability to integrate diverse biomarkers into a simplified medical algorithm implemented via a neuromorphic processor. By overcoming key challenges such as mechanical flexibility, signal variability, and scalability limitations, CSPINS offers a transformative solution for real-time, accessible, and low-cost diagnostic tools. This scalable, low-cost approach not only enhances usability but also opens pathways for addressing other complex medical conditions, positioning CSPINS as a versatile platform for advancing wearable healthcare technologies.”*

Comment 6. The authors should provide a full evaluation of the power consumption of CSPINS. On this basis, they should provide and demonstrate the use of wearable or conformable power unit and validate the functions of the whole system.

Additionally, they should benchmark the performance of the presented system respect to the metrics of a biological circuit performing similar functions. Which is the biology-to-

neuromorphic power consumption gap? How can this be overcome with future design/materials combinations?

Reply: We appreciate the reviewer's request for a comprehensive evaluation of CSPINS power consumption and a comparison with biological circuits. In response, we conducted a detailed analysis of power consumption of CSPINS and incorporated additional experimental findings into the revised manuscript.

For the synaptic node, the power consumption per signal exceeding the threshold is 0.14 nJ (**Fig. R6**). For the synapse devices, power consumption was measured as 2.2 μ J per write pulse and 2.5 μ J per erase pulse (**Fig. R7**). The proposed CSPINS system integrates four synapse devices, with each synapse operating based on six pulse inputs. As a result, a single CSPINS processing operation consumes approximately 112.8 μ J (37.6 μ W) within the processor with synaptic connections.

This value is higher compared to biological neural networks¹, and further optimization is needed through miniaturization of synapse devices and implementation of faster operation speeds. However, this is an expected trade-off given the inkjet-printed fabrication approach, which imposes constraints on device integration density and operating frequency synchronization with biomedical signals (e.g., heart rate). Despite this, CSPINS still demonstrates significantly lower power consumption compared to conventional wireless wearable data-processing systems, which typically require power in the mW range^{2,3}.

Our system is not intended to directly compete with biological neurons or other synaptic devices in terms of absolute power efficiency. Instead, CSPINS aims to provide an energy-efficient and compact alternative to conventional wearable electronics for real-time signal processing. Unlike traditional systems that rely on external data processing units and wireless communication, CSPINS integrates sensing and processing within the device, significantly reducing continuous data transmission requirements.

Even when considering the entire multimodal sensor system, including multiple physiochemical sensors, the overall power consumption remains in the few-mW level (**Note R1**), which is still substantially lower than that of conventional wearables healthcare systems requiring external data-processing resources and wireless communication systems. These results highlight the potential of CSPINS in enabling energy-efficient, real-time processing for wearable applications while maintaining a compact and scalable architecture.

1. Lee, Y., Park, H.-L., Kim, Y. & Lee, T.-W. Organic electronic synapses with low energy consumption. *Joule* **5**, 794–810 (2021).
2. Gao, W. *et al.* Fully integrated wearable sensor arrays for multiplexed in situ perspiration analysis. *Nature* **529**, 509–514 (2016).
3. Choi, Y. S. *et al.* A transient, closed-loop network of wireless, body-integrated devices for autonomous electrotherapy. *Science* **376**, 1006–1012 (2022).

Fig. R6 | Estimated energy consumption of the synaptic node. Real-time output changes of the synaptic node before and after the threshold and estimated power consumption.

Fig. R7 | Estimated energy consumption of the synapse. **a,b**, Real-time PSC changes of the synapse under repeated write operations (**a**) and the corresponding source-gate current plot and estimated power consumption during write operations (**b**). **c,d**, Real-time PSC changes of the synapse under repeated erase operations (**c**) and the corresponding source-gate current plot and estimated power consumption during the erase operations (**d**).

Note R1 | Evaluation of energy consumption of the entire CSPINS

To further discuss the competitive edge of CSPINS at the product level, we estimated the power dissipation of the connected sensors including the CBT sensor for core body temperature (CBT) calculation and the HR sensor for heart rate data processing.

Using the equation ' $P_{total} = P_1 + P_2 = I_2 \times (R_1 + R_2) = V_{in}^2 / (R_1 + R_2)$ ', we determined that the CBT sensor exhibited a power dissipation of 15.3 mW and the HR sensor exhibited a power dissipation of 12.5 mW. The CBT sensor operates with six pulses (pulse width: 0.05 s), while the HR sensor operates for 3 seconds. Based on these values, the power consumption of each was calculated to be 5.1 mJ for the CBT sensor and 5.3 mJ for the HR sensor. Therefore, the

total power consumption of CSPINS is estimated to be 15.7 mJ (5.23 mW) per processing operation.

Based on these findings, we have incorporated the following revisions into the manuscript:

In page 11: “Finally, the estimated power consumption of CSPINS was comprehensively evaluated in **Supplementary Fig. 37** and **Supplementary Table 7**. The power required for processing a single case was 37.6 μ W for synaptic operations and 5.3 mW when considering the entire system, demonstrating higher power efficiency compared to conventional medical diagnostic systems^{4,49}. Notably, even a small communication module in commercial wearable devices requires power in the mW range, further emphasizing the energy efficiency of CSPINS. These findings reinforce the feasibility of CSPINS for continuous, low-power health monitoring.”

In Supplementary Information:

Supplementary Table 7. | Evaluation of energy consumption of entire CSPINS

Component	Read current (μ A)	Operation voltage (V)	Resistance (k Ω)	Pulse number	Energy (μ J)	Power (μ W)
Chemical synapse	15.0	3.0	-	6.0	13.5	4.5
CBT (synapse)	15.0	3.4	-	6.0	15.3	5.1
HR (synapse)	15.0	3.5	-	6.0	15.8	5.3
CBT (sensor)	-	7.5	11.0	-	15340.9	5113.6
HR (sensor)	-	10.0	24.0	-	12500.0	4166.7
Synaptic node	0.02	0.15	-	1.00	0.0001	0.0008
Amplifier	-	10.0	3000.0	-	100.0	33.3

The power dissipation of the connected sensors including the CBT sensor for core body temperature (CBT) calculation and the HR sensor for heart rate data processing using the equation ‘ $P_{total} = P_1 + P_2 = I_2 \times (R_1 + R_2) = V_{in}^2 / (R_1 + R_2)$ ’.

In references: “49. Choi, Y. S. *et al.* A transient, closed-loop network of wireless, body-integrated devices for autonomous electrotherapy. *Science* **376**, 1006–1012 (2022).”

Comment 7. In Figure 6e, a clear flow chart is presented with and without von-Neumann architectures, to explain the function of the artificial “neurons” in parallel with the digital processing represented by AND and NOT gates. As such, what is the benefit of utilizing “neurons” in this case as a clear system can be defined (based of binary logic) and the “neurons” serve no function other than to integrate inputs? If the “neurons” are simply acting as threshold devices could this algorithm be implemented using another (simpler) circuitry? If this is to improve power consumption, what is the power consumption of the system, and would it not be lower removing either the von-Neumann component or the neurons to enable more efficient architectures? A key advantage of neuromorphic systems is the ability of the system to modulate/learn over time and on an event-driven basis. As such, are the thresholds (red and blue in Fig.5) determined by circuit design? This would limit the circuit leading to ineffective “training”, in which case, what training is done? For example, if the threshold are fixed by materials/circuit design, what is the point of this circuit since for example the level corresponding to “exercise” would be classified at level 1 with SIRS regardless of sepsis risk, since there is no motion information or even temporal information about the sensors?

Reply: We sincerely appreciate the reviewer's insightful comments and detailed questions. The points raised are highly relevant and have been carefully considered in our revised manuscript.

In this study, the synaptic nodes function as threshold-based processors rather than biologically inspired neuron. While the system could theoretically be implemented using conventional CMOS-based digital components such as Schmitt trigger, comparators etc., our approach offers distinct advantages:

1. Tunable threshold levels (**Fig. R5**) enabled by inkjet-printed fabrication, allowing mechanical flexibility and scalability compared to rigid CMOS circuits.
2. Simplified integration into wearable devices, reducing the need for external signal processing units and enabling compact, on-device computing.
3. Direct and intuitive implementation of diagnostic algorithms, reducing computational overhead for real-time biosignal analysis.

We acknowledge the reviewer's concern regarding the lack of learning capabilities in the current work, as neuromorphic systems typically exhibit adaptive features. While our device was primarily designed for efficient, real-time monitoring of hospitalized patients or individuals at risk of sepsis, we recognize its broader potential for wearable applications. Expanding its functionality for wearable use would require incorporating motion artifact correction, physical activity differentiation, and temporal information integration to enable dynamic response analysis and further enhance diagnostic accuracy. For instance, future iterations could incorporate additional sensors to compensate for motion-induced variations or implement advanced signal processing techniques to enhance classification accuracy without the need for extra hardware.

Currently, our synaptic nodes rely on fixed threshold values, but we envision the potential for incorporating dynamic threshold tuning via electrical control in future studies. This could enable adaptive behavior, improving diagnostic accuracy over time.

To address these considerations, we have updated the manuscript to emphasize both the current strengths of our synaptic node-based system and the potential advancements possible through future research.

In page 9: *“Importantly, the synapse-node integrated circuit demonstrated excellent mechanical flexibility and electrical durability, ensuring reliable performance in wearable environments (**Supplementary Figs. 31 and 32**). Our synaptic nodes function as threshold-based processors, offering greater adaptability and suitability for wearable devices compared to conventional complementary metal-oxide semiconductor (CMOS) circuits. This design enables for the seamless integration of diagnostic algorithms, representing a key advancement in developing hardware neural networks capable of analog signal processing for wearable neuromorphic computing applications.”*

In page 11: “It is important to note that while our current system relies on fixed threshold values, future iterations could incorporate adaptive threshold tuning via electrical control, enabling dynamic response adjustments over time. Additionally, integrating motion recognition and temporal information processing could further enhance diagnostic accuracy, ensuring robust and reliable classification even in real-world, ambulatory settings.”

Comment 8. Figure 6G: What is the accuracy presented here? Please include a confusion matrix, as well as enough normal samples for significance.

Reply: Thank you for your insightful comment. To address the reviewer’s concern regarding the accuracy and significance of normal samples, we conducted additional experiments by incorporating 11 normal cases into our dataset. These normal cases were selected from patients with common underlying conditions such as hypertension and diabetes, which are frequently observed in clinical settings (**Tables R1** and **R2**). This approach ensures that our model is evaluated in a realistic scenario, where patients exhibit baseline physiological variations without developing SIRS or sepsis.

The updated confusion matrix is shown in **Fig. R8**. From this confusion matrix, the overall accuracy of the CSPINS was estimated as 84.4%, distinguishing normal and SIRS with 100% classification while primary source of misclassification occurs between sepsis and shock cases. This demonstrates the CSPINS system's robustness and reliability for real-world clinical applications.

Table R1 | List of selected confusion cases for device validation. Biomarker data were extracted from patients diagnosed with hypertension (high blood pressure, HBP), diabetes (DM) or both.

Case No	Subject ID	Temp (°C)	HR (BPM)	Lactate (mM)	Diagnosis
22	124	37.0	77	1.3	HBP
23	188	36.6	79	4.8	HBP, DM
24	199	37.5	78	1.4	DM
25	209	36.6	73	0.9	HBP, DM
26	222	36.0	79	2.3	HBP
27	236	36.4	100	1.8	HBP
28	518	38.3	82	1.4	DM
29	533	37.2	69	1.3	DM
30	890	35.4	102	3.5	DM
31	1767	36.8	78	2.3	DM
32	2090	37.5	82	2.9	HBP, DM

Table R2 | Synaptic current and node outputs for various confusion case inputs. Simulation results of wearable neural network, detailing synaptic current response and corresponding node outputs.

Case No	Temp-Synapse	HR-Synapse	Lac-Synapse	SIRS	Sepsis	Shock	Normal
22	16.574	14.918	7.773	0	0	0	1
23	16.206	15.256	20.281	0	0	0	1
24	17.053	15.087	8.131	0	0	0	1
25	16.206	14.242	6.344	0	0	0	1
26	15.678	15.256	11.347	0	0	0	1
27	16.027	18.805	9.560	0	0	0	1
28	17.865	15.763	8.131	0	0	0	1
29	16.763	13.566	7.773	0	0	0	1
30	15.175	19.143	15.636	0	0	0	1
31	16.389	15.087	11.347	0	0	0	1
32	17.053	15.763	13.491	0	0	0	1

Fig. R8 | Confusion matrix for CSPINS validation. Confusion matrix summarizing the CSPINS system validation across 32 cases, including 6 SIRS, 7 SEPSIS, and 6 SEPTIC SHOCK cases.

We have carefully addressed these considerations in the revised manuscript.

In page 11: “Additionally, after incorporating 11 common non-inflammatory cases, such as hypertension and diabetes, we conducted a diagnostic simulation (**Supplementary Tables 5 and 6**). The results confirmed an overall accuracy of 84.4%, as evidenced by the confusion matrix (**Supplementary Fig. 36**).”

In supplementary information:

Supplementary Fig. 36 | Confusion matrix for CSPINS validation. Confusion matrix summarizing the CSPINS system validation across 32 cases, including 6 SIRS, 7 SEPSIS, and 6 SEPTIC SHOCK cases.

Supplementary Table 5. | List of selected confusion cases for device validation. Biomarker data were extracted from patients diagnosed with hypertension (high blood pressure, HBP), diabetes (DM) or both.

Case No	Subject ID	Temp (°C)	HR (BPM)	Lactate (mM)	Diagnosis
22	124	37.0	77	1.3	HBP
23	188	36.6	79	4.8	HBP, DM
24	199	37.5	78	1.4	DM
25	209	36.6	73	0.9	HBP, DM
26	222	36.0	79	2.3	HBP
27	236	36.4	100	1.8	HBP
28	518	38.3	82	1.4	DM
29	533	37.2	69	1.3	DM
30	890	35.4	102	3.5	DM
31	1767	36.8	78	2.3	DM
32	2090	37.5	82	2.9	HBP, DM

Supplementary Table 6. | Synaptic current and node outputs for various confusion case inputs. Simulation results of wearable neural network, detailing synaptic current response and corresponding node outputs.

Case No	Temp-Synapse	HR-Synapse	Lac-Synapse	SIRS	Sepsis	Shock	Normal
22	16.574	14.918	7.773	0	0	0	1
23	16.206	15.256	20.281	0	0	0	1
24	17.053	15.087	8.131	0	0	0	1
25	16.206	14.242	6.344	0	0	0	1
26	15.678	15.256	11.347	0	0	0	1
27	16.027	18.805	9.560	0	0	0	1
28	17.865	15.763	8.131	0	0	0	1
29	16.763	13.566	7.773	0	0	0	1
30	15.175	19.143	15.636	0	0	0	1
31	16.389	15.087	11.347	0	0	0	1
32	17.053	15.763	13.491	0	0	0	1

REVIEWER COMMENTS

We thank both reviewers for carefully reading our manuscript. We have revised the manuscript to address the remaining comments. The revised text is highlighted in red in the main text and below is a response to each comment.

Reviewer #1

The authors have successfully addressed my comments. The manuscript can now be accepted.

Reply: We thank the reviewer for carefully reading our paper and providing us with insightful and constructive feedback to further improve our paper.

Reviewer #3

The reviewer acknowledges that the other comments have been well addressed; however, there is still an issue regarding power consumption. The authors claimed that the system demonstrates advantages in terms of power consumption. However, throughout the manuscript, there is no supporting data comparing the power or energy consumption of this system with that of conventional CMOS-based systems. Please include a comparison table for power consumption in the manuscript, and provide a detailed explanation of the calculation processes.

Reply: We thank the reviewer for providing insightful and constructive comments. In response to this point, we have included a comparative table (**Table R1**) in the revised manuscript that summarizes representative wearable sensing and processing devices, along with their target biomarkers and reported power consumption values (as new **Supplementary Table 8**).

Page 11: “Finally, the estimated power consumption of CSPINS was comprehensively evaluated in **Supplementary Fig. 37** and **Supplementary Tables 7** and **8**. The power required for processing a single case was $37.6 \mu W$ for synaptic operations and $5.3 mW$ when considering the entire system, demonstrating higher power efficiency compared to conventional medical diagnostic systems^{4,49}.”

Table R1 | Comparison of the power consumption of CSPINS with previously reported wearable sensor-processor integration systems targeting various biomarkers.

Reference	Target biomarkers	Power Consumption
This work	Lactate, Core-body temperature, Heartrate	~ 5.3 mW
1	Glucose, Lactate, Na ⁺ , K ⁺	~ 92 mW
2	Glucose, Lactate, pH, Temperature	~ 55 mW
3	Alcohol , Glucose	~ 36 mW
4	Temperature, Heartrate, Motion	~ 274 mW
5	Heartrate	~ 400 mW

1. Gao, W. *et al.* Fully integrated wearable sensor arrays for multiplexed in situ perspiration analysis. *Nature* **529**, 509–514 (2016).
2. Choi, Y. S. *et al.* A transient, closed-loop network of wireless, body-integrated devices for autonomous electrotherapy. *Science* **376**, 1006–1012 (2022).
3. Kim, J. *et al.* Simultaneous Monitoring of Sweat and Interstitial Fluid Using a Single Wearable Biosensor Platform. *Adv. Sci.* **5**, 1800880 (2018).

4. Li, H., Sun, G., Li, Y. & Yang, R. Wearable Wireless Physiological Monitoring System Based on Multi-Sensor. *Electronics* **10**, (2021).
5. Majumder, A. J. A., ElSaadany, Y. A., Young Jr., R. & Ucci, D. R. An Energy Efficient Wearable Smart IoT System to Predict Cardiac Arrest. *Adv. Hum.-Comput. Interact.* **2019**, 1507465 (2019).